# What are you sinking? A geometric approach on attention sink

**Valeria Ruscio, Umberto Nanni, Fabrizio Silvestri**
Sapienza University of Rome
ruscio@diag.uniroma1.it, fsilvestri@diag.uniroma1.it

## Abstract

*Attention sink* (AS) is a consistent pattern in transformer attention maps where certain tokens (often special tokens or positional anchors) disproportionately attract attention from other tokens. We show that in transformers, AS is not an architectural artifact, but it is the manifestation of a fundamental geometric principle: the establishment of reference frames that anchor representational spaces. We analyze several architectures and identify three distinct reference frame types, centralized, distributed, and bidirectional, that correlate with the attention sink phenomenon. We show that they emerge during the earliest stages of training as optimal solutions to the problem of establishing stable coordinate systems in high-dimensional spaces. We show the influence of architecture components, particularly position encoding implementations, on the specific type of reference frame. This perspective transforms our understanding of transformer attention mechanisms and provides insights for both architecture design and the relationship with AS.

## 1 Introduction

Transformer-based models exhibit an interesting phenomenon called "attention sink" where beginning-of-sequence tokens receive substantial attention (30-40%) regardless of semantic relevance [Xiao et al., 2023]. Removing this pattern degrades model performance, suggesting these allocations serve an essential function beyond content processing. We postulate that attention sinks represent transformers' reference frames, coordinate systems within their representational manifolds. Transformer operations rely on dot products between query and key vectors, making angular relationships crucial for information flow. Reference frames provide fixed geometric anchors that allow tokens to establish relative positions through consistent angular relationships, solving the challenge of maintaining stable geometric relationships in high-dimensional spaces. Without these anchors, token representations would lack consistent orientation, making reliable encoding of positional and semantic relationships impossible. We show that reference frames emerge as mathematically optimal solutions to constraints imposed by the softmax operation on the probability simplex. We categorize them into three classes: centralized, distributed, and bidirectional. While prior work [Gu et al., 2024, Barbero et al., 2025] treated attention sinks as model-specific phenomena, our geometric interpretation unifies these observations as alternative solutions to the same fundamental challenge, advancing understanding of transformer geometry.

## 2 Related works

The attention sink phenomenon, where tokens allocate substantial attention to specific tokens regardless of semantic relevance, was first identified by Xiao et al. [2023], who discovered the beginning-of-sequence token consistently receives disproportionate attention. This aligns with broader efforts to develop mechanistic understanding of transformer models [Elhage et al., 2021]. While Yu et al.

39th Conference on Neural Information Processing Systems (NeurIPS 2025).

[2024] and Cancedda [2024] found attention sinks emerging beyond sequence beginnings (which our framework explains as distributed reference frames), Gu et al. [2024] observed that high cosine similarity between queries and the first token's keys creates attention sinks despite the keys' small $\ell_2$-norm, a phenomenon our theory reframes as the low-norm keys establishing a distinguished point in the representation manifold. Zhang et al. [2025]'s "catch, tag, and release" mechanism aligns with our finding that reference frames emerge early in training as mathematical necessities, while Barbero et al. [2025] frames attention sinks as preventing information 'over-mixing' in deep networks.

Position encoding significantly influences attention patterns: Su et al. [2024]'s Rotary Position Embedding creates bias toward first-token attention (facilitating centralized reference frames), while alternatives like ALiBi [Press et al., 2021] demonstrate how encoding changes affect attention patterns. Ruscio and Silvestri [2024] found wavelet-like patterns emerging among attention heads to overcome RoPE limitations, and Kazemnejad et al. [2023] showed scaled RoPE variants dramatically change attention patterns, which our framework explains as enabling distributed reference frames through reduced positional bias. Research by Liu et al. [2024], Voita et al. [2019], and Mohebbi et al. [2023] supports our prediction that architectural design directly influences reference frame formation. Finally, Darcet et al. [2024] observed that absolute position embeddings create topological complexity supporting bidirectional reference frames (consistent with our encoder-only findings), while Sun et al. [2024] demonstrated that explicit bias parameters can mitigate attention sinks, which our framework interprets as altering the manifold's geometric structure.

# 3   Reference Frames

It has been shown by Moschella et al. [2022] that in high-dimensional representation spaces, latent vectors need a consistent way to relate to each other by establishing reference frames acting as canonical coordinate systems to anchor the representation manifold. Without stable reference points, the model struggles to consistently determine relationships between tokens, and distances and directions become ambiguous. At its core, a reference frame in a transformer is a structure $\mathcal{R} = (\mathcal{M}, \mathcal{P}, \phi)$ where $\mathcal{M}$ is the representation manifold (a smooth, locally Euclidean topological space), $\mathcal{P} = \{p_1, p_2, ..., p_k\}$ is a set of reference points that act as distinguished locations on the manifold, and $\phi : \mathcal{M} \times \mathcal{P} \to \mathbb{R}^d$ is a mapping function that relates any point to the reference points, providing a coordinate chart for the manifold. This structure provides a means to consistently measure relationships between token representations regardless of their semantic content or position.

This definition allows us to formalize what constitutes an "attention sink" mathematically: an attention sink is a token position $j$ for which the attention weight $\alpha_{ij}$ exceeds a threshold $\tau$ across many source tokens $i$:

$$\text{sink}(j) = \left[ \frac{1}{n} \sum_{i=1}^{n} \mathbb{1}_{\{\alpha_{ij} \geq \tau\}} \right] \geq \gamma \tag{1}$$

where $\mathbb{1}$ is the indicator function, $\tau$ is typically set to the 90th percentile of attention weights, and $\gamma$ is a frequency threshold (typically 0.3-0.5).

Reference frames emerge through self-organization during training rather than being explicitly programmed. However, this self-organization occurs within architecture-specific channels that significantly influence the resulting geometric structures. This guided emergence can be formalized as optimizing a loss function $\mathcal{L}$ over an architecture-specific inductive bias $\mathcal{B}$.

These inductive biases don't deterministically program specific attention patterns but rather create the conditions where certain geometric structures naturally emerge as optimal solutions during training. The architecture shapes the loss landscape such that gradient descent naturally converges toward specific reference frame types. This explains why attention sinks consistently form during training without being explicitly encoded in the architecture.

## 3.1   Vector geometry of reference frame types

We identify three distinct reference frame types, each characterized by a specific geometric organization within the attention mechanism's vector space. Our analysis shows that attention sinks tokens consistently receive disproportionate attention weight regardless of their semantic content, functioning as geometric anchors in the representation space. These attention sinks fall into two distinct categories: they appear either as special tokens (like [BOS] or [CLS] and [SEP]) that mark

sequence boundaries or as regular vocabulary tokens (such as commas or articles) that serve syntactic functions in the text.

**Centralized Reference Frames** emerge in decoder-only architectures with standard RoPE (LLaMA 3.1 and 3.2, Mistral v0.1, Gemma). The beginning-of-sequence token [BOS] becomes a universal origin point where each token's query vector maintains high cosine similarity with this reference token's key vector, even as the reference key's magnitude ($\ell_2$-norm) remains small. This creates a computational hub where all tokens can efficiently establish their relative positions through a single comparison operation. When processing "The cat chased the mouse", attention from "mouse" would focus substantially (30-40%) on the [BOS] token rather than semantically related tokens, creating an efficient comparison path "mouse ↔ [BOS] ↔ cat". This can be expressed as a transformation where token representations are oriented primarily through their relationship to the central reference point:

$$\mathbf{h}'_i = \alpha_{i,\text{BOS}}\mathbf{v}_{\text{BOS}} + \sum_{j \neq \text{BOS}} \alpha_{ij}\mathbf{v}_j \tag{2}$$

where $\alpha_{i,\text{BOS}}$ typically ranges from 0.3-0.4 (30-40%) regardless of semantic relationship.

**Distributed Reference Frames** emerge in architectures with modified positional encoding schemes (Qwen 2.5, Phi-2). Multiple tokens serve as reference points, creating a more flexible coordinate system. At the vector level, multiple key vectors maintain moderate cosine similarity with various query vectors across the sequence, creating a network of local reference points. In our example, attention from "mouse" would be allocated (10-15%) to several anchoring tokens, creating multiple computational paths: "mouse ↔ [Ref] ↔ cat" and "mouse ↔ "the" ↔ cat." This distributed reference structure creates multiple, smaller-weight transformations:

$$\mathbf{h}'_i = \sum_{r \in \mathcal{R}} \alpha_{i,r}\mathbf{v}_r + \sum_{j \notin \mathcal{R}} \alpha_{ij}\mathbf{v}_j \tag{3}$$

where $\mathcal{R}$ is the set of reference tokens and $\alpha_{i,r}$ typically ranges from 0.1-0.15 (10-15%) for each reference.

**Bidirectional Reference Frames** emerge in encoder architectures with absolute position embeddings (BERT, XLM-RoBERTa). Both the beginning and end tokens serve as reference points, with attention patterns shifting through network depth. Early layers establish relationships with the beginning token's vector, while deeper layers increasingly reference the end token's vector, creating a dynamic coordinate system that changes through network depth. This dynamic reference structure can be formalized as a layer-dependent transformation:

$$\mathbf{h}_i^{(l)} = \sum_{j \in \{\text{start,end}\}} \beta_j^{(l)}\alpha_{ij}\mathbf{v}_j + \sum_{j \notin \{\text{start,end}\}} \alpha_{ij}\mathbf{v}_j \tag{4}$$

where $\beta_j^{(l)}$ are layer-specific weighting factors that shift from start-dominant in early layers to end-dominant in later layers.

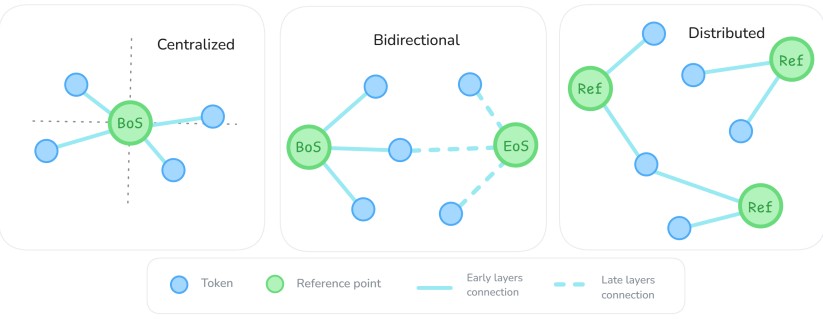

Figure 1: Geometric interpretation of reference frames: (left) *centralized frame* with a single dominant reference point serving as a universal origin; (center) *distributed frame* with multiple weaker reference points creating a flexible coordinate system; (right) *bidirectional frame* with a dual-anchor structure and layer-wise specialization.

When a token attends to a reference point, it performs a vector operation that orients its representation within the shared coordinate system:

$$\mathbf{h}'_i = \sum_j \alpha_{ij}(\mathbf{W}_V \mathbf{x}_j) \approx \alpha_{i,\text{ref}}(\mathbf{W}_V \mathbf{x}_{\text{ref}}) + \sum_{j \neq \text{ref}} \alpha_{ij}(\mathbf{W}_V \mathbf{x}_j) \quad (5)$$

This explains why removing attention sinks often disrupts performance despite their seeming semantic irrelevance, they provide the geometric infrastructure that makes consistent representation possible. The mathematical significance of reference frames becomes clear in the eigendecomposition of attention matrices. Dominant eigenvectors align with reference tokens, creating stable subspaces that serve as coordinate axes. This alignment is captured in the low-rank approximation $A \approx UV^\top$, where columns of $U$ correspond to reference points that minimize the Frobenius norm error $|A - UV^\top|_F$. These reference structures enable transformers to perform implicit basis transformation operations that maintain geometric consistency across sequence positions.

## 3.2 Information Geometry and the Probability Simplex

The softmax operation in attention is fundamental to reference frame formation, as it creates a geometric constraint through the probability simplex [Zuhri et al., 2025]:

$$\Delta^{n-1} = \{(a_1, a_2, ..., a_n) \in \mathbb{R}^n \mid a_j \geq 0 \text{ for all } j, \text{ and } \sum_j a_j = 1\} \quad (6)$$

This constraint mathematically necessitates the emergence of reference frames by creating two critical geometric effects: i) it transforms unbounded attention logits into a bounded manifold with intrinsic curvature, and ii) it enforces a conservation law that makes attention a zero-sum resource, more about it in A. The resulting geometry privileges sparse attention distributions that concentrate probability mass on a minimal set of tokens, precisely the pattern observed in reference frames. Each attention distribution $\mathbf{a}_i$ represents a point on the probability simplex $\Delta^{n-1}$, inducing a Riemannian metric through the Fisher information matrix $\mathcal{F}_{ij} = \mathbb{E}[\partial_i \log p(x) \partial_j \log p(x)]$. This metric determines how distances are measured in the representation space and creates the conditions for attention sinks to emerge as geodesic reference points.

In the context of attention mechanisms, the information bottleneck principle [Tishby et al., 2000] translates to optimizing attention distributions on the probability simplex:

$$\min_{a_1,...,a_n \in \Delta^{n-1}} \sum_{i,j} w_{ij} D_{KL}(a_i || a_j) + \lambda R(a_1, ..., a_n) \quad (7)$$

where $w_{ij}$ represents the semantic relevance between tokens $i$ and $j$, and $R$ is a regularization term. The attention distribution from each token defines a probability measure on the manifold, inducing a metric structure through the Fisher information matrix. This metric determines how distances are measured in the representation space.

## 3.3 The Emergence of Attention Sinks

The attention sink phenomenon can be formalized through the interaction between position encoding and the self-attention mechanism. In self-attention, the attention score between tokens at positions $i$ and $j$ is:

$$\alpha_{ij} = \frac{\exp(\mathbf{q}_i \cdot \mathbf{k}_j / \sqrt{d})}{\sum_{l=1}^n \exp(\mathbf{q}_i \cdot \mathbf{k}_l / \sqrt{d})} \quad (8)$$

where $\mathbf{q}_i = \mathbf{W}_Q \mathbf{x}_i$ and $\mathbf{k}_j = \mathbf{W}_K \mathbf{x}_j$ are the query and key vectors. With RoPE, this becomes:

$$\alpha_{ij} = \frac{\exp((\mathbf{R}_{\theta_i} \mathbf{W}_Q \mathbf{x}_i) \cdot (\mathbf{R}_{\theta_j} \mathbf{W}_K \mathbf{x}_j) / \sqrt{d})}{\sum_{l=1}^n \exp((\mathbf{R}_{\theta_i} \mathbf{W}_Q \mathbf{x}_i) \cdot (\mathbf{R}_{\theta_l} \mathbf{W}_K \mathbf{x}_l) / \sqrt{d})} \quad (9)$$

For the first token ($j = 1$), $\mathbf{R}_{\theta_1} = \mathbf{I}$ (the identity matrix), creating a computational advantage that biases attention toward this position. This mathematical property, combined with the simplex constraint of the softmax, naturally leads to the formation of an attention sink at the beginning-of-sequence token in models with standard RoPE. Specifically, when $\mathbf{k}_1$ has a small $\ell_2$-norm relative to other keys but maintains high angular alignment with queries, it creates the conditions for an attention

sink. This can be expressed through the dot product decomposition: $\mathbf{q}_i \cdot \mathbf{k}_1 = |\mathbf{q}_i| \cdot |\mathbf{k}_1| \cos(\theta_{q_i,k_1})$. This balance between small norm and high cosine similarity represents an optimal trade-off in the attention mechanism's geometry. A small $|\mathbf{k}_1|$ ensures the reference token doesn't dominate the output representation magnitudes, while high $\cos(\theta_{q_i,k_1})$ ensures consistent geometric relationships. This explanation clarifies why attention sinks aren't merely artifacts but optimal geometric solutions that emerge consistently during training despite different initialization conditions.

### 3.4 Position Encoding and Reference Frame Formation

Position encoding implementations fundamentally shape the geometric organization of transformer attention, directly influencing which reference frame type emerges during training. Our analysis shows thatthe mathematical structure of position encoding creates specific inductive biases that guide attention patterns toward distinct geometric configurations. Different encoding schemes establish different coordinate geometries through their modifications of token embeddings:

**Standard Rotary Position Embeddings (RoPE)** used in architectures like LLaMA, apply a rotation matrix $\mathbf{R}_\theta$ with frequency-dependent angles: $\mathbf{x}_i \mapsto \mathbf{R}_{\theta_i}\mathbf{x}_i$ where $\theta_i = i \cdot \omega$ for fixed frequency $\omega$. This creates a fundamental geometric asymmetry: the first token (position 0) receives no rotation at all ($\theta_0 = 0$), making its rotation matrix the identity matrix ($\mathbf{R}_0 = \mathbf{I}$). This mathematical property gives the first token a privileged computational status in the attention mechanism. As other tokens undergo increasingly larger rotations based on their positions, their query vectors maintain higher cosine similarity with the first token's key vector compared to other tokens at similar semantic distances. During training, the model naturally exploits this computational advantage, developing a centralized reference frame where the first token becomes a universal origin point in the representation space. Topological analysis of these models reveals star-like attention graphs with a dominant central node, reflected in low $\text{Betti}_1$ values (few cycles) and strong negative correlation between algebraic connectivity and degree centralization. This structure corresponds to a *pointed manifold* $(\mathcal{M}, p_0)$ with a distinguished point $p_0$ serving as a universal origin. This geometric configuration optimizes the probability simplex constraint by placing high probability mass on a single token, creating a sparse attention distribution that enables efficient long-range dependency modeling.

**NTK-aware scaled RoPE** [1] employed by models such as Qwen 2.5 and Phi-2, introduces a critical geometric modification to standard rotary embeddings. By applying a scaling factor $\alpha < 1$ to the rotation angles: $\mathbf{x}_i \mapsto \mathbf{R}_{\alpha \cdot \theta_i}\mathbf{x}_i$. This approach fundamentally alters the manifold geometry of attention. The reduced rate of angular separation diminishes the computational advantage of the first token by making the relative rotations between tokens more uniform throughout the sequence. This seemingly minor mathematical change has profound effects on reference frame formation. With the positional bias toward the first token weakened, the model no longer converges on a single dominant reference point during training. Instead, multiple tokens can effectively compete as reference points, creating a distributed network of local coordinate systems. Spectral analysis of these models reveals a characteristic sign-flipping correlation pattern between connectivity and centralization metrics that reflects this fundamental reorganization of the attention manifold. The geometric transformation can be understood as a shift from a pointed manifold $(\mathcal{M}, p_0)$ with a single distinguished origin to a multi-pointed manifold $(\mathcal{M}, \{p_1, p_2, ..., p_n\})$. This distributed structure sacrifices some of the efficient hub-and-spoke compression of centralized frames for greater contextual adaptability, enabling more flexible modeling of diverse linguistic structures.

**Absolute position embeddings** implemented in encoder architectures like BERT and XLM-RoBERTa, employ a fundamentally different approach to encoding position: $\mathbf{x}_i \mapsto \mathbf{x}_i + \mathbf{p}_i$, where $\mathbf{p}_i$ is a learned embedding for position $i$. Unlike rotational methods that modify the geometric relationships between tokens through angular transformations, absolute embeddings directly inject position-specific information into each token's representation. This creates a different kind of inductive bias, rather than establishing implicit reference points through computational advantages, absolute embeddings create explicit position markers throughout the sequence. This approach naturally supports the emergence of bidirectional reference frames with reference points at both

---

[1]NTK stands for Neural Tangent Kernel, in the context of transformers, this approach modifies position encodings based on network capacity rather than using fixed frequencies, allowing models to better generalize to lengths beyond their training distribution.

sequence boundaries. Topologically, these models display remarkably high initial complexity with elevated $\text{Betti}_1$ values, indicating numerous cycles in early layers, a strong contrast to the star-like structures in decoder models. The spectral analysis shows peak connectivity in early but not initial layers, with dramatic connectivity drops between early and middle layers, suggesting rapid geometric reorganization. As the network deepens, attention systematically shifts between reference points, creating a dynamic coordinate system that changes through network depth. This bipolar manifold $(\mathcal{M}, p_{\text{start}}, p_{\text{end}})$ implements layer-specific optimization strategies that shift attention mass between special tokens, enabling simultaneous access to both sequence endpoints, a critical capability for tasks requiring bidirectional context integration.

## 4   Methodology

**Geometric and Topological Analysis**   Standard attention analysis cannot distinguish structured coordinate systems from semantic similarity patterns. We quantified the topological structure of attention networks through persistent homology and Betti numbers, revealing how connectivity patterns evolve across network depth. Our approach draws on the growing application of topological data analysis to neural networks [Naitzat et al., 2020, Moor et al., 2020] and the established methodology of persistent homology for analyzing high-dimensional data structures [Edelsbrunner et al., 2008]: (Distance Matrix $= 1 -$ Attention Matrix).
The Ripser algorithm computed persistent homology on these distance matrices, tracking $\text{Betti}_0$ (connected components) and $\text{Betti}_1$ (cycles) across layers. Persistence values measure the significance of topological features, with higher values indicating more stable structures. This approach revealed characteristic topological signatures for each reference frame type.
To complement topological features with algebraic characterization, we examined the spectral properties of attention graphs' Laplacian matrices to assess connectivity patterns and reference structures ($L = D - A$ where $D$ is the degree matrix and $A$ is the adjacency matrix derived from thresholded attention weights). We computed these matrices at multiple thresholds (0.001, 0.005, 0.01, 0.02, 0.05, 0.1, 0.2), measuring: i) Algebraic connectivity (Fiedler value) to quantify graph connectedness; ii) Star-likeness to measure proximity to ideal star topologies, directly testing for hub-based reference frames; iii) Gini coefficient to quantify inequality in attention distribution; and iv) Degree centralization to assess concentration of attention. We analyzed correlations between these metrics to identify mathematical signatures of different reference frame types.

**Information-Theoretic Analysis**   While topology reveals structural patterns, understanding whether reference tokens merely aggregate information or fundamentally establish coordinate systems requires quantifying their functional impact. We first quantified information-geometric properties of attention distributions, particularly how attention sinks affect information flow: KL Reduction $=$ KL(original) $-$ KL(without sinks). Our implementation identified attention sinks using percentile thresholds (0.8, 0.9, 0.95), measuring the changes in KL divergence when attention sinks were removed, the attention sink concentration across network layers, and the layer-specific patterns in how reference points influence information geometry. Large KL reductions indicate that sinks establish organizational principles rather than superficial attention patterns.
We then examined how reference frames manifest in value space geometry through complementary quantitative approaches. We measure the influence of reference tokens by calculating their relative magnitude, the ratio between reference token keys' $\ell_2$-norm and average keys ($\|\mathbf{k}_{\text{ref}}\|_2 / \frac{1}{n} \sum_i \|\mathbf{k}_i\|_2$), revealing how these points establish distinguished positions in representation space. We capture directional guidance through cosine similarity between reference values and transformation vectors ($\frac{1}{n} \sum_{i=1}^{n} \cos(\mathbf{v}_{\text{ref}}, \mathbf{h}_i' - \mathbf{h}_i)$), while quantifying their structural importance via KL divergence between original and reference-removed attention matrices ($D_{\text{KL}}(A \| A_{\text{ref}})$). To distinguish between reference frame types, we identify the number of significant reference points across architectures. We further investigate the relationship between attention and geometric transformations by examining attention entropy ($- \sum_j a_{ij} \log a_{ij}$), transformation magnitude ($\|\mathbf{h}_i' - \mathbf{h}_i\|_2$), and their correlation ($\text{corr}(H(A_i), \|\mathbf{h}_i' - \mathbf{h}_i\|_2)$). We also measure geometric-semantic alignment ($\text{corr}(a_{ij}, \cos(\mathbf{v}_i, \mathbf{v}_j))$) to determine whether attention follows semantic relationships or prioritizes geometric organization. The directional influence metric is particularly interesting as it directly quantifies how reference tokens shape the geometric structure of the manifold. High values indicate that reference tokens actively guide the orientation of all other token representations, functioning as coordinate axes rather than merely aggregating contextual information. Similarly, geometric-semantic alignment measures

whether attention patterns follow semantic content (positive values) or abstract geometric principles (negative values), providing insight into how reference frames balance content processing with coordinate system establishment.

To understand how reference frames affect learning dynamics and parameter sensitivity, we computed the Fisher information matrix, which provides a natural Riemannian metric on the manifold of probability distributions, measuring how sensitive model outputs are to parameter perturbations. For each model, we computed the layer-wise Fisher norm $\|F_\ell\|_F = \sqrt{\sum_{i,j}[F_\ell]_{ij}^2}$ where $F_\ell$ represents the Fisher information matrix for parameters in layer $\ell$. This quantifies the information content and learning capacity of each layer with respect to the attention distribution manifold, revealing whether reference frame layers exhibit distinctive information-geometric properties.

**Random Matrix Theory Analysis of Attention Evolution**    The temporal emergence of reference frames shows whether they represent fundamental architectural necessities or learned optimizations. If reference structures emerge early and consistently across initializations, they likely constitute essential computational primitives. To study this, we employed Random Matrix Theory (RMT) to analyze how attention structures emerge from initially random patterns: $p_{MP}(x) = \frac{\sqrt{(b-x)(x-a)}}{2\pi\gamma x}, \quad a \leq x \leq b$ where $a = (1 - \sqrt{\gamma})^2$, $b = (1 + \sqrt{\gamma})^2$, and $\gamma$ is the aspect ratio of the matrix. Deviation from this Marchenko-Pastur distribution indicates the emergence of non-random structure. We quantified this emergence through several metrics: Spectral Gap $= \frac{\lambda_1}{\lambda_2}$, Participation Ratio $= \frac{(\sum_i \lambda_i)^2}{\sum_i \lambda_i^2}$

$$D_{KL}(p_{emp}||p_{MP}) = \sum_i p_{emp}(i) \log \frac{p_{emp}(i)}{p_{MP}(i)} \tag{10}$$

where $\lambda_i$ are the eigenvalues of the attention matrix, and $p_{emp}$ is the empirical eigenvalue distribution. For each checkpoint during training, we extracted attention matrices from all layers and heads, computed their eigendecomposition, and tracked the evolution of these metrics. This methodology allows us to identify when reference frames begin to form and how they develop through training, providing insights into the fundamental role these structures play in the learning process.

We tracked these metrics across training checkpoints of Pythia models from the earliest stages (step0 to step8) to later points (step9000 through step143000), identifying when reference frames begin to form and how they develop through training.

**Experimental design**    We analyzed a diverse set of transformer models to investigate architecture-specific and architecture-invariant patterns in reference frame formation: **decoder-only models** - LLaMA-3.2 (1B, 3B) and 3.1 (8B-Instruct, 8B), Phi-2, Qwen-2.5 (3B, 7B, 7B-Instruct), Mistral-7B-v0.1, Gemma-7B, Pythia (1.4B, 2.8B, 6.9B, 12B); and **encoder-only models** - BERT-base-uncased, XLM-RoBERTa-large.

For topological, spectral graph, value space and KL divergence analyses, we used a dataset of STEM-focused Wikipedia sentences (mathematics, chemistry, medicine, physics) ranging from 6 to 50 tokens. We processed 500 samples for topology, spectral and Fisher information analysis, and a subset of 50 samples for KL divergence analysis. For the temporal RMT analysis, we examined 100 samples across training checkpoints of Pythia models. All experiments were conducted using Google Colab with T4 or A100 GPUs.

## 5    Analysis Results

As we can see from table 1 each type exhibits characteristic mathematical signatures across our analytical methods, yet all establish stable coordinate systems for representation learning.

### 5.1    Centralized Reference Frames

Centralized reference frames emerge in decoder-only architectures with standard rotary position embeddings (RoPE), including LLaMA, Mistral, and Gemma. Our spectral graph analysis reveals strong negative correlation between algebraic connectivity (Fiedler value) and degree centralization, indicating that centralized attention structures prioritize information concentration over distributed connectivity, as shown in table 2. This creates a star-like topology with a dominant central node. The KL divergence in table 3 analysis consistently shows negative KL reduction values when attention

Table 1: Topological Analysis of Reference Frame Types

| | Property | Llama-3.2-3B | Qwen2.5-7B | XLM-RoBERTa |
|---|---|---|---|---|
| | **Reference frame type** | Centralized | Distributed | Bidirectional |
| Connected Components | Early layer ($Betti_0$) | 26.43 | 26.16 | 22.68 |
| | Final layer ($Betti_0$) | 26.43 | 17.97 | 1.69 |
| | Change | 0.00 | -8.19 | -20.99 |
| Cycles/Loops | Early layer ($Betti_1$) | 0.00 | 0.00 | 19.69 |
| | Final layer ($Betti_1$) | 0.00 | 0.00 | 3.11 |
| | Change | 0.00 | 0.00 | -16.58 |
| Topological Persistence | Early layer persistence | 0.0573 | 0.2381 | 0.0521 |
| | Final layer persistence | 0.1620 | 0.1543 | 0.0156 |
| | Change | +0.1046 | -0.0838 | -0.0365 |
| Attention Head Specialization | Token specialization | 100% on BoT | 65.4% on "," | 100% on / |
| | Specialized heads | 120 | 7 | 77 |
| | Top layer specialization | Layer 0 (24 heads) | Layer 0 (7 heads) | Layer 18 (16 heads) |
| Attention Standard Deviation | Early layer | 0.1219 | 0.0737 | 0.0197 |
| | Middle layer (max) | 0.1620 (layer 21) | 0.1316 (layer 14) | 0.0950 (layer 12) |
| | Final layer | 0.1332 | 0.1182 | 0.0602 |

sinks are removed, demonstrating the reference point's critical role in maintaining geometric stability. Fisher information metrics in table 8 show extreme early-layer concentration, with approximately 60% of total Fisher information concentrated in the first layers, quantifying precisely how these architectures establish their coordinate system through a dominant early reference point. The full results for LLaMa E, Mistral v0.1 and Gemma G and Pythia J are in the appendix.

Table 2: Spectral graph signatures of reference frame types

| | Property | Centralized (LLaMA-3.2-3B) | Distributed (Qwen2.5-7B) | Bidirectional (RoBERTa) |
|---|---|---|---|---|
| Algebraic Connectivity (Fiedler Value) | Position encoding | Standard RoPE | NTK-aware RoPE | Absolute |
| | High threshold effectiveness | 0.04 (1/28 layers) | 0.25 (7/28 layers) | 0.38 (9/24 layers) |
| | Early / Middle / Late layers | 12.5 / 11.7 / 7.0 | 16.2 / 12.8 / 11.0 | 38.3 / 31.2 / 22.5 |
| | Maximum value (layer) | 17.7 (0) | 18.8 (1) | 55.3 (1) |
| Star-likeness Measure | Low thrsh. (0.001): E/M/L | 0.54 / 0.53 / 0.54 | 0.54 / 0.53 / 0.50 | 0.35 / 0.36 / 0.39 |
| | High thrsh. (0.1): E/M/L | 0.97 / 0.96 / 0.93 | 0.78 / 0.93 / 0.88 | 0.85 / 0.95 / 0.94 |
| | Middle-layer peak (high thrs.) | No | Yes | Yes |
| Degree Centralization and Variance | Centralization: E/M/L | 0.54 / 0.55 / 0.61 | 0.52 / 0.54 / 0.50 | 0.08 / 0.13 / 0.22 |
| | Variance: E/M/L | 114.9 / 111.6 / 85.1 | 130.3 / 117.4 / 104.5 | 98.3 / 50.8 / 89.7 |
| Signature Correlation Patterns | Fiedler vs. Centraliz. (0.001) | Strong negative (-0.95) | Negative (-0.46) | Positive (0.32) |
| | Fiedler vs. Centraliz. (0.1) | Positive (0.34) | Strong positive (0.61) | Negative (-0.33) |
| | Correlation sign flip | Yes (-0.95 → +0.34) | Yes (-0.46 → +0.61) | Yes (0.32 → -0.33) |

## 5.2 Distributed Reference Frames

Distributed reference frames emerge in architectures with modified positional encoding schemes, such as Qwen 2.5 and Phi-2s NTK-aware scaled RoPE. Our spectral analysis identifies a distinctive sign-flipping correlation pattern between Fiedler values and centralization metrics, negative at low thresholds transitioning to positive at higher thresholds, shown in table 2, a reliable signature of distributed reference frames. KL divergence measurements in table 3 reveal a characteristic three-phase pattern: positive KL reduction in early layers, stronger negative reduction in middle layers, and moderated negative values in late layers. This signature indicates that reference points serve different functions at different network depths. Fisher information in table 8 shows lower peak concentration with multiple significant peaks across different network depths, reflecting a fundamentally different approach to establishing coordinate systems. The complete results for Qwen 2.5 F and Phi-2 I are in the appendix.

Table 3: Architectural Differences in Attention Sink Properties

|  | Property | LLaMA-3.2-3B | Qwen2.5-7B | XLM-RoBERTa |
|---|---|---|---|---|
|  | Reference frame type | Centralized | Distributed | Bidirectional |
| Attention Sink Properties (t=0.8) | Avg. KL Reduction | -0.0974 | -0.0088 | -0.1017 |
|  | Avg. Sink Concentration | 82.93% | 69.92% | 66.86% |
|  | Max Sink Concentration | 96.40% (L25) | 85.75% (L23) | 85.22% (L21) |
| Layer-wise Distribution | Early Layer Pattern | Strong negative KL | Positive KL | Mixed KL |
|  | Middle Layer Pattern | Moderate negative KL | Mixed KL | Strong negative KL |
|  | Deep Layer Pattern | Strong negative KL | Mixed KL | Variable KL |
| Architectural Implications | Sink Formation | Consistent across layers | Variable by layer | Strong middle-layer focus |
|  | Threshold Sensitivity | High | Moderate | Moderate |
|  | Early-layer Context | Strong sink focus | Weak sink formation | Very weak sink formation |

## 5.3 Bidirectional Reference Frames

Bidirectional reference frames emerge in encoder architectures with absolute position embeddings, such as BERT and XLM-RoBERTa. Our topological analysis reveals remarkably high initial complexity with high $Betti_1$ values indicating numerous loops/cycles in early layers, contrasting sharply with decoder models, as shown in table 1. The spectral analysis in table 2 shows peak connectivity in early but not initial layers, with dramatic drops in connectivity from early to middle layers suggesting rapid geometric reorganization. The most distinctive signature appears in our KL divergence analysis, in table 3, which reveals a characteristic U-shaped profile with positive KL reduction in both first and final layers, confirming the dual-anchor nature of the reference structure. Fisher information in table 8 peaks in middle layers rather than at the beginning, indicating a fundamentally different approach to information distribution. The comprehensive results for BERT and XML-RoBERTa H are in the appendix.

## 5.4 Value space analysis

Our vector geometry analyses reveal how reference frames function as coordinate systems within transformer representation manifolds. As shown in Table 4, each reference frame type implements a distinct geometric strategy for balancing representational stability with flexibility. Centralized frames

Table 4: Value Space Characteristics Across Reference Frame Types

|  | Property | LLaMA-3.2-3B | Qwen2.5-7B | XLM-RoBERTa |
|---|---|---|---|---|
|  | Reference frame type | Centralized | Distributed | Bidirectional |
| Directional Influence | First Layer | 0.9672 | 0.7508 | 0.7310 |
|  | Middle Layer | 0.5058 | 0.5000 | 0.9808 |
|  | Last Layer | 0.5012 | 0.5000 | 0.9274 |
|  | Evolution Pattern | Sharp decrease | Early decrease, then flat | Increase, then stable |
| Geometric-Semantic Alignment | First Layer | -0.0383 | 0.0569 | 0.2065 |
|  | Middle Layer | -0.2999 | -0.0771 | -0.3545 |
|  | Last Layer | -0.2614 | 0.0000 | -0.3982 |
|  | Evolution Pattern | Consistently negative | Near zero throughout | Positive to strongly negative |
| Information Content Change | Mean | 410.97 | 7161.51 | 13.76 |
|  | Std Dev | 126.99 | 3649.72 | 4.40 |
|  | Pattern | Moderate | Very high | Low |
| Reference Token Structure | Mean Count | 1.00 | 1.23 | 1.60 |
|  | Maximum Count | 2.00 | 8.00 | 10.00 |
|  | Distribution | Single token dominant | Multiple tokens | Varying tokens by layer |
| Attention-Value Relationships | Entropy-Magnitude Corr. | 0.23 | -0.23 | 0.23 |
|  | Early-to-Late Layer Shift | -0.03 to -0.57 | -0.70 to -0.05 | -0.16 to 0.64 |
|  | Transformation Magnitude | 2.46 to 84.07 | 15.03 to 222.39 | 19.74 to 23.81 |

establish a strong initial coordinate origin that gradually accommodates more nuanced transformations while maintaining a single reference point. The negative geometric-semantic alignment confirms these frames prioritize geometric organization over semantic relationships. Distributed frames employ multiple reference points with substantially higher information content change, resembling differential geometry's use of local coordinate charts for complex manifolds. This approach trades computational efficiency for greater adaptability to semantic structure. Bidirectional frames exhibit a remarkable phase transition in geometric-semantic alignment, implementing a dual-phase computation where

early layers leverage semantic relationships before deeper layers reorganize representations according to more abstract geometric principles.

These patterns confirm that reference frames represent optimized solutions to the challenge of establishing stable coordinate systems in high-dimensional spaces, with architectural choices directly influencing which strategy emerges.

Our Random Matrix Theory analysis further supports this interpretation by demonstrating that reference frames emerge during the earliest training steps, well before task performance begins to converge, indicating their fundamental role in representation organization.

## 5.5 Temporal Emergence of Reference Frames

Our Random Matrix Theory analysis in table 5 shows how reference frames develop during the earliest stages of training across different model scales. The spectral gap metric shows a non-monotonic relationship with model size, smaller models exhibit minimal changes while mid-sized models (particularly 6.9B) demonstrate the most pronounced increases. This suggests reference frames establish themselves most efficiently at certain parameter scales rather than scaling linearly with model size. Participation ratio measurements show that while smaller models develop more distributed attention during training, larger models progressively concentrate attention in fewer dimensions. The dramatic increase in participation ratio change between 6.9B and 12B models suggests a phase transition in how the largest models organize their representational geometry.

Table 5: Evolution of Random Matrix Theory Properties During Pythia Model Training

|  | Metric | Pythia-1.4B | Pythia-2.8B | Pythia-6.9B | Pythia-12B |
|---|---|---|---|---|---|
| Average Changes | Spectral Gap | -0.0002 | 0.0014 | 0.0031 | 0.0007 |
| from Step 0 | Participation Ratio | 0.0073 | -0.0111 | -0.0118 | -0.0727 |
| to Step 8 | Attention Entropy | 0.0005 | 0.0003 | 0.0011 | 0.0018 |
|  | Sink Concentration | 0.0001 | 0.0002 | 0.0001 | 0.0013 |
| Layer-wise | Largest Spectral Gap Increase | 0.0163 (L18) | 0.0139 (L28) | 0.0288 (L30) | 0.0322 (L23) |
| Distribution | Largest Spectral Gap Decrease | -0.0177 (L17) | -0.0138 (L31) | -0.0250 (L23) | -0.0227 (L35) |
| of Changes | Largest Part. Ratio Increase | 0.1530 (L13) | 0.1792 (L19) | 0.4147 (L23) | 0.3061 (L25) |
|  | Largest Part. Ratio Decrease | -0.1353 (L17) | -0.2898 (L23) | -0.4957 (L25) | -0.8785 (L34) |

Layer specialization becomes increasingly pronounced as models scale up, with early layers remaining relatively stable, middle layers showing divergent patterns, and deep layers demonstrating dramatic evolution in attention structure. The dual trends in attention entropy and sink concentration metrics reveal that while attention generally becomes more uniformly distributed across tokens, larger models simultaneously develop more pronounced attention sinks. The emergence of increasingly structured attention patterns in larger models suggests that scale enables more sophisticated geometric representations, a mathematical necessity rather than an architectural accident.

# 6    Limitations and Conclusion

**Limitations**    Our topological and spectral analyses focus on attention patterns at specific network snapshots rather than continuously tracking their evolution throughout training.

**Conclusion**    Our work reframes attention sinks from architectural quirks to fundamental aspects of transformer geometry. The reference frame perspective offers three key contributions: (1) unifying diverse attention patterns across architectures through a single geometric principle; (2) providing insights into how architectural choices influence attention organization; and (3) establishing a foundation for deliberate reference frame engineering. By demonstrating that attention patterns reflect optimal solutions to the challenge of establishing stable coordinate systems in high-dimensional spaces, our framework opens new directions for transformer optimization. Future works will explore using attention sink tokens as anchoring points for transfer learning, potentially enabling more efficient knowledge transfer while preserving geometric stability across different model architectures.

# 7 Acknowledgments

Research supported in part by European Union - Next Generation EU - namely by the MUR-PRIN 2022 project "2022REWNTE - Artificial Intelligence algorithms to track and detect Covid-19 vaccine-related infodemic on social media" - CUP no.B53D23020690006; in part by the projects FAIR under Grant PE0000013 and SERICS under Grant PE00000014 under the MUR National Recovery and Resilience Plan funded by the European Union-NextGenerationEU, and in part by the project Neural Reasoning over Open Data (NEREO) funded by the Italian Ministry of Education and Research (PRIN) under Grant 2022AEFHA, and in part by the project SEED funded by Sapienza University of Rome.

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

## A Geometric Effects of the Softmax Operation

The softmax operation transforms raw attention logits into probability distributions through the equation $a_j = \exp(z_j)/\sum_i \exp(z_i)$. This transformation has two fundamental geometric consequences that directly drive reference frame formation.

First, softmax transforms unbounded attention logits into a bounded manifold with intrinsic curvature. While input logits $z_j$ can range from $-\infty$ to $+\infty$ in Euclidean space $\mathbb{R}^n$, the output attention weights $a_j$ are confined to the probability simplex $\Delta^{n-1}$. This mapping from Euclidean space to the simplex introduces non-Euclidean geometry with positive curvature under the Fisher-Rao metric. The curvature of this manifold means that geodesics (shortest paths) differ from straight lines, creating geometric pressure toward the vertices and edges of the simplex. Mathematically, this corresponds to sparse attention distributions where most probability mass concentrates on a small subset of tokens. The dimensional reduction from $n$-dimensional logit space to an $(n-1)$-dimensional simplex creates an information bottleneck that forces the network to prioritize certain geometric relationships over others during training.

Second, softmax enforces a conservation law that makes attention a zero-sum resource through the constraint $\sum_j a_j = 1$. This constraint means each token has exactly one unit of total attention to distribute across all tokens in the sequence. Any increase in attention weight to one token must be precisely balanced by decreases to others, creating an economy of attention where tokens compete for finite resources. This competition creates evolutionary pressure during training toward allocation patterns that maximize the computational utility of each attention unit. The simplex constraint drives the model toward solutions that balance the trade-off between distributing attention for content processing and concentrating attention for geometric reference. The mathematical optimality of allocating substantial attention to a small set of reference tokens emerges naturally from this constrained optimization problem.

Together, these geometric effects create the necessary conditions for reference frames to emerge as optimal solutions to the challenge of establishing stable coordinate systems while respecting the mathematical constraints imposed by the softmax operation. The different reference frame types we identify represent alternative solutions to this same fundamental geometric problem, each balancing the trade-offs between computational efficiency and representational flexibility in mathematically distinct ways.

# B Statistical Analysis

Our statistical methodology employed multiple complementary approaches to validate the existence of distinct reference frame types. We performed pairwise t-tests between transformer layers to identify significant changes in topological metrics (Betti numbers and persistence values), using $\alpha = 0.05$ as our significance threshold. For spectral analysis, we computed Pearson and Spearman correlations between Fiedler values and graph properties (star-likeness, centralization, variance, density) across seven attention thresholds (0.001 to 0.2), allowing us to detect threshold-dependent correlation patterns. To quantify information-theoretic differences, we calculated KL divergence between original and sink-removed attention distributions, testing the statistical significance of KL reduction values and their correlation with sink concentration metrics. Fisher information analysis involved computing Frobenius norms of Fisher matrices across layers and correlating these with layer depth using both Pearson and Spearman methods to capture linear and monotonic relationships respectively. All correlation analyses included p-value calculations to assess statistical significance, with $p < 0.05$ considered significant. When comparing multiple metrics across numerous layers, we report the proportion of significant comparisons to evaluate overall pattern reliability.

Table 6: Core Statistical Metrics Across Reference Frame Types

| | Property | LLaMA-3.2-3B | Qwen2.5-7B | XLM-RoBERTa |
|---|---|---|---|---|
| | Reference frame type | Centralized | Distributed | Bidirectional |
| Topological Stability | $Betti_0$ constancy | $26.41 (p = 1.000)$ | $26.13 (p = 1.000)$ | $22.69 \rightarrow 1.00 (p < 0.0001)$ |
| | Dim0 Persistence range | $0.032 - 0.162 (p < 0.0001)$ | $0.067 - 0.418 (p < 0.0001)$ | $0.003 - 0.155 (p < 0.0001)$ |
| | $Betti_1$ evolution | 0 (constant) | 0 (constant) | $20.93 \rightarrow 0 (p < 0.0001)$ |
| Spectral Correlations (Fiedler) | Star-likeness (t=0.001) | $r = -0.32, p = 0.10$ | $r = -0.59, p < 0.001$ | $r = -0.76, p = 0.008$ |
| | Star-likeness (t=0.02) | $r = 0.44, p = 0.016$ | $r = 0.51, p < 0.001$ | $r = 0.57, p = 0.004$ |
| | Centralization (t=0.001) | $r = -0.48, p < 0.001$ | $r = -0.44, p = 0.008$ | $r = 0.31, p = 0.052$ |
| | Centralization (t=0.02) | $r = -0.57, p < 0.001$ | $r = -0.63, p = 0.031$ | $r = -0.68, p = 0.038$ |
| Information Metrics | KL reduction range | $-0.274 to + 0.013$ | $-0.219 to + 0.110$ | $-0.284 to + 0.088$ |
| | KL-concentration correlation | $r = -0.78, p < 0.001$ | $r = -0.76, p = 0.0015$ | $r = -0.70, p = 0.008$ |
| | Peak sink concentration | 96.06% | 86.46% | 85.52% |
| Learning Dynamics | Fisher-depth correlation | $r = -0.39, p = 0.042$ | $r = -0.79, p < 0.001$ | $r = -0.38, p = 0.071$ |
| | Fisher-depth Spearman | $\rho = -0.97, p < 0.001$ | $\rho = -0.82, p < 0.001$ | $\rho = -0.38, p = 0.066$ |
| | Attention parameters (%) | 12.7% | 35.4% | 72.8% |

## B.1 Reference frames statistical analysis

In table 6 is our comprehensive statistical analysis provides quantitative validation for the three distinct reference frame types identified in transformer architectures. Centralized reference frames (LLaMA-3.2-3B) exhibit remarkable topological invariance with $Betti_0$ remaining constant at 26.41 across all layers (p = 1.000) and zero $Betti_1$ values throughout, while Dim0 Persistence evolves significantly from 0.057 in early layers to 0.162 in the final layer (all p < 0.0001), accompanied by spectral correlation reversals where Fiedler-star correlations shift from negative (r = -0.32) at low thresholds to positive (r = 0.50, p < 0.001) at higher thresholds, consistently negative KL reduction values across layers (ranging from -0.03 to -0.274) with strong negative correlation between KL reduction and sink concentration (r = -0.78, p < 0.001), and a significant negative Fisher information correlation with layer depth (r = -0.39, p = 0.042) showing extremely strong monotonic decrease (Spearman $\rho$ = -0.97, p < 0.001).

Distributed reference frames (Qwen2.5-7B) demonstrate a distinctive U-shaped Dim0 Persistence pattern dropping from 0.418 to 0.067 before recovering to 0.261 (all transitions p < 0.0001), systematic spectral correlation reversals across thresholds where star-likeness correlations change from negative (r = -0.59, p < 0.001) to positive (r = 0.51, p < 0.05) while centralization correlations strengthen negatively, a three-phase KL reduction pattern with positive values in early layers (+0.110), negative in middle layers (-0.219), and mixed in deep layers, alongside the strongest Fisher information depth correlation (r = -0.79, p < 0.001) indicating efficient learning distribution across multiple reference points.

Bidirectional reference frames (XLM-RoBERTa) uniquely show dramatic topological evolution with $Betti_0$ decreasing from 22.69 to 1.00 and $Betti_1$ from 20.93 to near-zero (all p < 0.0001), layer-specific spectral behaviors varying from strong negative star-likeness correlations in early layers (r = -0.92, p < 0.001) to positive correlations in deeper layers, a characteristic U-shaped KL profile with positive

reduction at extremes (+0.071 and +0.088) and negative values in middle layers (minimum -0.28), combined with no significant Fisher information correlation with depth (r = -0.38, p = 0.071) reflecting maintained learning capacity throughout the network, where attention parameters comprise 72.8% of total parameters compared to 12.7% in centralized and 35.4% in distributed frames, demonstrating how each reference frame type implements a mathematically distinct solution to coordinate system establishment in high-dimensional transformer representations.

## B.2 Value space statistical analysis

Table 7: Value space statistical analisis. The asterisks indicate statistical significance levels (* $p < 0.05$, **$p < 0.01$, *** $p < 0.001$).

| | Property | LLaMA-3.2-3B | Qwen2.5-7B | XLM-RoBERTa |
|---|---|---|---|---|
| | **Reference frame type** | Centralized | Distributed | Bidirectional |
| Reference Point Metrics | Relative Magnitude (mean) | 0.5413 | 0.4893 | 0.5918 |
| | Directional Influence (mean) | 0.5419 | 0.5523 | 0.9397 |
| | Information Content Change | 411.18 | 7084.96 | 13.74 |
| | Reference Token Count | 1.00 | 1.23 | 1.58 |
| | Max Reference Count | 2 | 7 | 9 |
| Layer Evolution (Correlations) | Relative Magnitude Pattern | Mixed ($r = -0.37$) | Mixed ($r = 0.29$) | Mixed ($r = 0.33$) |
| | Directional Influence Pattern | Mixed ($r = -0.37$) | Decreasing ($r = -0.60$) | Mixed ($r = -0.46$) |
| | Information Content Pattern | Mixed ($r = 0.46$) | Increasing ($r = 0.53$) | Mixed ($r = -0.03$) |
| Cross-Layer Differences | Directional Influence Change (Early vs Late) | $-0.082$ ($p = 0.091$) | $-0.105*$ ($p = 0.045$) | $-0.028$ ($p = 0.290$) |
| | Information Content Change (Early vs Late) | $+119.66*$ ($p = 0.014$) | $+2452.50$ ($p = 0.061$) | $-0.024$ ($p = 0.990$) |
| Key Reference Correlations | Magnitude-Influence Correlation | $r = 0.9997***$ ($p < 0.001$) | $r = -0.1264***$ ($p < 0.001$) | r=-0.0018 ($p = 0.843$) |
| | Magnitude-Info Content Correlation | $r = -0.9147***$ ($p < 0.001$) | $r = 0.2285***$ ($p < 0.001$) | $r = 0.6000***$ ($p < 0.001$) |
| | Influence-Info Content Correlation | $r = -0.9153***$ ($p < 0.001$) | $r = -0.7708***$ ($p < 0.001$) | $r = 0.6268***$ ($p < 0.001$) |
| Attention-Value Relationships | Attention Entropy (mean) | 1.0522 | 1.4513 | 2.1871 |
| | Value Transform Magnitude | 9.7099 | 16.1056 | 8.1544 |
| | Geometric-Semantic Alignment | $-0.2923***$ ($p < 0.001$) | $-0.0405***$ ($p < 0.001$) | $0.0019$ ($p = 0.561$) |
| | Entropy-Magnitude Correlation | $0.2279***$ ($p < 0.001$) | $-0.2294***$ ($p < 0.001$) | $0.2346***$ ($p < 0.001$) |
| Layer-wise Evolution | Attention Entropy Change (First to Last) | $-0.4068***$ ($p < 0.001$) | $-2.1086***$ ($p < 0.001$) | $-1.0198***$ ($p < 0.001$) |
| | Transform Magnitude Change (First to Last) | $+81.6071***$ ($p < 0.001$) | $+206.3281***$ ($p < 0.001$) | $+4.0779***$ ($p < 0.001$) |
| | Geometric Alignment Change (First to Last) | $-0.2224***$ ($p < 0.001$) | $-0.0575***$ ($p < 0.001$) | $-0.6038***$ ($p < 0.001$) |
| Statistical Properties | Sample Size (texts) | 500 | 500 | 500 |
| | Total Data Points | $14,000$ | $14,000$ | $12,000$ |
| | Significant Correlations | $37/40 (92.5\%)$ | $36/40 (90.0\%)$ | $33/40 (82.5\%)$ |

Our statistical analysis of value space provides quantitative validation for the three distinct reference frame types, with measurements across 500 text samples generating $12,000 - 14,000$ data points per architecture. Centralized reference frames (LLaMA-3.2-3B) demonstrate the most coherent reference structure with near-perfect correlation between relative magnitude and directional influence ($r = 0.9997, p < 0.001$), maintaining a single dominant reference token (mean count $= 1.0006$) whose directional influence dramatically decreases from 0.97 in the first layer to 0.50 in the final layer, while information content change significantly increases from early (351.35) to late layers (471.01) with a mean difference of 119.66 ($t = -2.76, p = 0.014$), accompanied by consistently negative geometric-semantic alignment (mean $= -0.29, p < 0.001$) and positive entropy-magnitude correlation ($r = 0.23, p < 0.001$) indicating that diverse attention patterns produce larger value transformations. Distributed reference frames (Qwen2.5-7B) employ multiple reference points (mean count $= 1.23$, max $= 7$) with substantially higher information content change (mean $= 7084.96$, SD $= 3641.75$), showing a distinctive negative correlation between relative magnitude and directional influence ($r = -0.13, p < 0.001$) that contrasts sharply with centralized frames, while directional influence significantly decreases from early (0.60) to late layers (0.50) with $p = 0.045$, accompanied by near-zero geometric-semantic alignment (mean $= -0.04, p < 0.001$) and negative entropy-

magnitude correlation ($r = -0.23$, $p < 0.001$) suggesting that focused attention patterns drive larger transformations in this architecture. Bidirectional reference frames (XLM-RoBERTa) maintain the highest overall directional influence (mean $= 0.94$, SD$= 0.07$) with the most complex reference structure (mean count $= 1.58$, max$= 9$), uniquely showing no correlation between relative magnitude and directional influence ($r = -0.002$, $p = 0.84$) while both metrics correlate positively with information content change ($r = 0.60$ and $r = 0.63$, both $p < 0.001$), exhibiting the only non-significant geometric-semantic alignment (mean $= 0.002$, $p = 0.56$) and positive entropy-magnitude correlation ($r = 0.23$, $p < 0.001$). Cross-layer evolution reveals dramatic and statistically significant changes across all architectures: attention entropy decreases substantially (centralized: $-0.41$, distributed: $-2.11$, bidirectional:$-1.02$, all $p < 0.001$), value transformation magnitude increases exponentially (centralized: 81.61-fold increase from 2.46 to 84.07, distributed: 206.33-fold from 15.03 to 222.39, bidirectional: 4.08-fold from 19.74 to 23.81, all $p < 0.001$), while geometric-semantic alignment becomes increasingly negative (centralized: $-0.22$ decrease, distributed: $-0.06$ decrease, bidirectional: $-0.60$ decrease, all $p < 0.001$), with layer-specific correlation patterns showing remarkable consistency within each architecture type, centralized frames exhibit correlation reversals from negative ($r = -0.53$) to positive ($r = -0.57$) entropy-magnitude relationships, distributed frames show systematic progression from strongly negative ($r = -0.70$) to near-zero ($r = -0.05$) correlations, and bidirectional frames demonstrate the most dramatic shifts with correlations ranging from $r = -0.16$ to $r = 0.64$ across layers. These quantitative patterns provide robust statistical evidence that reference frames not only organize attention patterns but fundamentally shape value space transformations, with each architecture implementing mathematically distinct strategies that manifest in measurable differences in how information flows through the network, how strongly reference points influence transformations, and how semantic and geometric properties interact throughout the model depth.

## C   Fisher Table

Table 8: Comparative Analysis of Fisher Information Distribution Across LLaMa, Qwen and RoBERTa

|  | **Property** | **LLaMA-3.2-3B** | **Qwen2.5-7B** | **XLM-RoBERTa** |
|---|---|---|---|---|
|  | **Reference frame type** | Centralized | Distributed | Bidirectional |
| Component Importance | Attention Mechanism | 12.7% (78,164) | 29.7% (46,040) | 71.7% (8,363,327) |
|  | MLP Components | 85.5% (527,103) | 51.8% (80,380) | 27.8% (3,244,160) |
|  | Embedding | 1.8% (10,892) | 2.1% (3,247) | 0.2% (26,120) |
| Layer Distribution | Peak Layer | Layer 1: 371,465 | Layer 0: 18,791 | Layer 9: 2,504,416 |
|  | Secondary Peak | Layer 0: 39,429 | Layer 26-27: 6,039 | Layer 8: 2,371,845 |
|  | Peak to Minimum Ratio | 143:1 | 26:1 | 303:1 |
| Structural Features | Total Fisher Norm | 616,694 | 155,080 | 11,668,807 |
|  | Early Layers (0-9) Concentration | 82.7% | 64.7% | 59.3% |
|  | Middle Layers Concentration | 13.5% | 14.9% | 39.8% |
|  | Final Layers Concentration | 3.8% | 3.9% | 0.9% |

## D   Attention Sink Data

The analysis of attention sink patterns across 500 Wikipedia samples provides strong evidence for the theoretical predictions regarding positional embedding geometry. Llama 3.2 3B, employing standard rotary position embeddings (RoPE), shows the predicted pointed manifold structure with the beginning-of-text token serving as the dominant attention sink in 100% of analyzed samples, achieving an average sink score of 0.9968. This universal reliance on a single reference point directly reflects the geometric asymmetry introduced by RoPE, where the first token experiences no rotation ($\theta_0 = 0$, $\mathbf{R}_0 = \mathbf{I}$), giving a privileged computational status that the model exploits during training. In stark contrast, Qwen 2.5 3B, which implements NTK-aware scaled RoPE with reduced angular separation ($\alpha < 1$), has a fundamentally different attention topology consistent with a multi-pointed manifold ($\mathcal{M}, \{p_1, p_2, ..., p_n\}$). Rather than converging on a universal origin, Qwen distributes attention sinks across diverse tokens—with the most common sink (*The*) appearing in only 17.2% of samples and the top 10 sinks cumulatively accounting for 53.4% of cases. This distributed

Table 9: Attention sink distribution summary across 500 Wikipedia samples. Analysis performed with $\gamma = 0.3$ (sink threshold) and $\tau$ set to the 90th percentile of attention weights.

| Model | Samples | Most Common Top Sink | Occurrence | % Coverage | Avg Sink Score |
|---|---|---|---|---|---|
| Llama 3.2 3B | 500 | `<|begin_of_text|>` | 500 | **100.0%** | 0.9968 |
| Qwen 2.5 3B | 500 | *The* | 86 | **17.2%** | 0.8892 |

Table 10: Distribution of top 10 attention sink tokens for Llama 3.2 3B (standard RoPE) and Qwen 2.5 3B (NTK-aware scaled RoPE). Llama exhibits extreme centralization with 100% coverage by a single token, while Qwen shows distributed sink selection across diverse tokens.

| Llama 3.2 3B (Standard RoPE) | | | | Qwen 2.5 3B (NTK-aware RoPE) | | | |
|---|---|---|---|---|---|---|---|
| Rank | Token | Freq. | % | Rank | Token | Freq. | % |
| 1 | `<|BOS|>` | 500 | 100.0 | 1 | *The* | 86 | 17.2 |
| 2 | *medicine* | 4 | 0.8 | 2 | *,* | 51 | 10.2 |
| 3 | *matter* | 3 | 0.6 | 3 | *In* | 40 | 8.0 |
| 4 | *include* | 3 | 0.6 | 4 | *This* | 19 | 3.8 |
| 5 | *MRI* | 2 | 0.4 | 5 | *For* | 17 | 3.4 |
| 6 | *pressure* | 2 | 0.4 | 6 | *It* | 15 | 3.0 |
| 7 | *logy* | 2 | 0.4 | 7 | *These* | 10 | 2.0 |
| 8 | *Nunes* | 2 | 0.4 | 8 | *of* | 10 | 2.0 |
| 9 | *GIS* | 2 | 0.4 | 9 | *A* | 10 | 2.0 |
| 10 | *conversion* | 1 | 0.2 | 10 | *One* | 9 | 1.8 |
| **Cumulative (Top 10):** | | | **101.2%** | **Cumulative (Top 10):** | | | **53.4%** |

pattern, where different contexts select different reference tokens, confirms that NTK-aware scaling successfully mitigates the positional bias inherent in standard RoPE, enabling the formation of multiple local coordinate systems rather than a single centralized reference frame.

# E Llama Family

As we can see from table 11, all Llama variants, regardless of parameter count or fine-tuning, exhibit 100% specialization on the beginning-of-sequence token. This is a strong evidence fot the centralized reference frame in this kind of architecture. The stable $Betti_0$ numbers (connected components) across all layers in all models confirm the predicted behavior of centralized reference frames, they maintain consistent topological structure rather than merging components as seen in distributed frames. All models show monotonically increasing persistence values from early to late layers, and this indicates that the centralized reference point becomes more significant in deeper layers, suggesting increasing reliance on this coordinate system as information flows through the network.

The remarkable consistency across different parameter scales (1B, 3B, 8B) suggests that reference frame formation is a fundamental architectural property rather than an emergent behavior dependent on model size. Despite some difference, the instruction-tuned model still maintains the fundamental centralized reference frame signature, perfect BOS specialization, stable topology, and monotonically increasing persistence.

In table 12 we can see that all Llama models exhibit remarkably similar spectral properties despite their varying parameter counts (1B to 8B) and fine-tuning status. This consistency supports the thesis that reference frame type is fundamentally determined by architectural choices rather than scale or training objective. The consistent peak in algebraic connectivity in middle layers (1.01×-1.61× higher) aligns with our theory that centralized reference frames optimize information flow through the network via a coordination layer structure. The strongest peak in the smallest model (1.61× in 1B vs. 1.01× in 8B) suggests that smaller models rely more heavily on this coordination mechanism.

The strong negative correlation between algebraic connectivity and degree centralization (-0.9790 to -0.9865) is a distinctive mathematical signature of centralized reference frames. This indicates that as

Table 11: Reference frame signatures across Llama model variants

| Property | Llama-3.2-1B | Llama-3.2-3B | Llama-3.1-8B | Llama-3.2-3B-Instruct |
|---|---|---|---|---|
| BOS specialization | 100% | 100% | 100% | 100% |
| $Betti_0$ (early) | 26.39 | 26.43 | 26.55 | 26.37 |
| $Betti_0$ (late) | 26.39 | 26.43 | 26.55 | 26.37 |
| $Betti_0$ change | 0.00 | 0.00 | 0.00 | 0.00 |
| Dim0 persistence (early) | 0.0748 | 0.0573 | 0.0591 | 0.0410 |
| Dim0 persistence (late) | 0.1651 | 0.1620 | 0.1612 | 0.1350 |
| Persistence increase | +0.0903 | +0.1046 | +0.1021 | +0.0940 |
| Max attention StdDev | 0.1400 | 0.1620 | 0.1635 | 0.1622 |

connectivity increases, attention becomes more evenly distributed, but critically, this happens only at specific thresholds, maintaining the overall centralized structure.

Table 12: Spectral Signatures in Llama models

| Property | Llama-3.2-1B | Llama-3.2-3B | Llama-3.2-3B-I | Llama-3.1-8B |
|---|---|---|---|---|
| Position encoding | RoPE | RoPE | RoPE | RoPE |
| Threshold effectiveness (0.1) | 1.00 | 0.04 | 0.04 | 0.03 |
| Algebraic connectivity (early) | 13.3843 | 12.4644 | 12.9546 | 12.6150 |
| Algebraic connectivity (middle) | 13.9363 | 11.6529 | 12.7520 | 10.7345 |
| Algebraic connectivity (late) | 8.5773 | 6.9879 | 8.3986 | 7.8740 |
| Maximum connectivity | 18.4033 (layer 0) | 17.6574 (layer 0) | 18.2137 (layer 0) | 18.0901 (layer 0) |
| Star-likeness (early) | 0.5348 | 0.5352 | 0.5368 | 0.5345 |
| Star-likeness (middle) | 0.5351 | 0.5340 | 0.5358 | 0.5327 |
| Star-likeness (late) | 0.5357 | 0.5371 | 0.5394 | 0.5343 |
| Maximum star-likeness | 0.5377 (layer 14) | 0.5517 (layer 22) | 0.5526 (layer 22) | 0.5455 (layer 25) |
| Degree centralization (early) | 0.5383 | 0.5435 | 0.5418 | 0.5436 |
| Degree centralization (middle) | 0.5315 | 0.5502 | 0.5424 | 0.5582 |
| Degree centralization (late) | 0.5796 | 0.6126 | 0.5909 | 0.5979 |
| Maximum centralization | 0.6169 (layer 13) | 0.6988 (layer 22) | 0.6720 (layer 22) | 0.6683 (layer 25) |
| Degree variance (early) | 119.9924 | 114.9265 | 119.7555 | 115.2534 |
| Degree variance (middle) | 124.9291 | 111.6136 | 118.7679 | 106.3977 |
| Degree variance (late) | 100.6093 | 85.1385 | 97.0134 | 89.3467 |

All models show high threshold sensitivity, with meaningful graph structures only emerging at lower thresholds ($\leq 0.05$). This sensitivity is itself a signature of centralized frames, where attention is concentrated on specific tokens, creating sparse attention distributions that become disconnected at higher thresholds. All models exhibit a sign-flip in correlation patterns at approximately the same threshold ($\approx 0.2$), transitioning from negative to positive correlation between Fiedler values and centralization metrics. This threshold uniformity across model scales suggests it represents a fundamental property of centralized reference frame organization rather than a scale-dependent phenomenon.

In table 14 we can see that unlike the three-phase pattern observed in Qwen's distributed reference frames, Llama models exhibit a consistent pattern of negative KL reduction values across nearly all layers, with only layer 0 showing small positive values. This consistent negative KL reduction pattern (averaging between -0.07 and -0.13 across different network regions) indicates that removing attention sinks in Llama models consistently reduces the uniformity of attention distributions. This provides strong evidence for the centralized reference frame hypothesis, where a single dominant reference point serves as a universal coordinate system for all token representations.

A critical finding is the extremely high sink concentration values across all network depths in Llama models (77-89%), substantially higher than the early-layer concentrations in Qwen models (39-48%). This indicates that Llama establishes strong reference points immediately and maintains them throughout the network.

Table 13: Key Fiedler value correlations across LLaMA models at different thresholds

| Threshold | Property | Mean Correlation | Pattern |
|---|---|---|---|
| 0.001 | Centralization vs. Fiedler | -0.6361 | Strong negative |
| 0.01 | Centralization vs. Fiedler | -0.8795 | Very strong negative |
| 0.02 | Centralization vs. Fiedler | -0.8621 | Very strong negative |
| 0.1 | Centralization vs. Fiedler | 0.4720 | Sign flip to positive |
| 0.001 | Density vs. Fiedler | 0.4808 | Moderate positive |
| 0.01 | Density vs. Fiedler | 0.8221 | Very strong positive |
| 0.1 | Density vs. Fiedler | 0.5293 | Moderate positive |

Several key patterns provide additional evidence for centralized reference frames: for example, the layer with maximum sink concentration is consistently deep in the network (layer 25 for three models), reaching extraordinary values of 96+%. This suggests the reference point becomes increasingly important for coordinate stabilization as representations grow more complex through network depth. The most negative KL reduction (strongest effect from removing sinks) occurs either in very early layers (L1) or very deep layers (L25), revealing a bimodal pattern where reference points are established early and then heavily leveraged in later processing. The base and instruction-tuned 3B models show remarkable similarity in their reference frame signatures, suggesting the centralized reference structure is fundamental to the architecture rather than task-dependent.

The consistency of these patterns across model scales (1B to 8B) demonstrates that centralized reference frames represent a stable architectural solution to the geometric organization problem.

Table 14: Attention Sink Analysis Across Llama Model Variants

| Property | Llama-3.1-8B | Llama-3.2-1B | Llama-3.2-3B | Llama-3.2-3B-Instruct |
|---|---|---|---|---|
| KL Reduction (0-3 layers) | -0.1093 | -0.0959 | -0.1271 | -0.1213 |
| KL Reduction (5-15 layers) | -0.0726 | -0.0741 | -0.0692 | -0.0803 |
| KL Reduction (17+ layers) | -0.1275 | N/A | -0.1208 | -0.1179 |
| Max KL reduction | 0.0118 (L0) | 0.0221 (L0) | 0.0072 (L0) | 0.0370 (L0) |
| Min KL reduction | -0.2245 (L25) | -0.2040 (L1) | -0.2175 (L1) | -0.2225 (L1) |
| Sink concentration (0-3 layers) | 77.09% | 75.91% | 83.29% | 83.77% |
| Sink concentration (5-15 layers) | 83.36% | 75.98% | 79.94% | 82.58% |
| Sink concentration (17+ layers) | 89.22% | N/A | 87.51% | 88.76% |
| Max sink concentration | 96.60% (L25) | 95.33% (L13) | 96.40% (L25) | 96.13% (L25) |
| Layer pattern type | Consistent | Consistent | Consistent | Consistent |
| Optimal threshold | 0.8 | 0.8 | 0.8 | 0.8 |
| Analyzed layers | 17 | 9 | 15 | 15 |

# F Qwen Family

As we can see in table 17, all Qwen models exhibit the distributed reference pattern, with moderate specialization (36.6%-65.4%) on common linguistic elements rather than the beginning-of-sequence token. This contrasts sharply with the centralized reference frames in Llama models where BOS specialization is consistently 100%. A fascinating pattern emerges in $Betti_0$ changes: the smaller 3B model maintains stable component counts ($26.18 \rightarrow 26.18$). While both 7B models show significant component integration, with similar reductions (-8.19 and -9.27) This suggests that larger models develop more flexible reference structures where components merge in deeper layers, potentially enabling more complex reasoning by integrating information across multiple reference points.

All Qwen models show decreasing persistence values through network depth and this confirms that distributed frames initially establish stronger reference points that gradually weaken as information flows through the network. The instruction-tuned model shows slightly enhanced component integration (-9.27) compared to its base counterpart (-8.19), suggesting that instruction tuning further optimizes the distributed reference mechanism for improved reasoning.

Table 15: Fisher information distribution across Llama models

| Architectural Pattern | LLaMA-3.2-1B | LLaMA-3.2-3B | LLaMA-3.2-3B-I | LLaMA-3.1-8B |
|---|---|---|---|---|
| **Component Importance (Percentage of Total Fisher Norm)** | | | | |
| Attention Mechanism | 11.0% | 12.7% | 23.3% | 47.9% |
| MLP Components | 86.9% | 85.5% | 70.1% | 30.2% |
| Embeddings | 2.1% | 1.8% | 6.4% | 20.7% |
| **Layer Distribution Patterns** | | | | |
| First Two Layers | 66.7% | 66.7% | 65.8% | 61.6% |
| Middle Layers (4-11) | 18.7% | 15.2% | 11.5% | 6.4% |
| Deep Layers (12+) | 12.3% | 18.1% | 22.6% | 10.7% |
| **Layer Decay Characteristics** | | | | |
| Layer 1 to Layer 2 Ratio | 19.4:1 | 19.2:1 | 25.8:1 | 9.5:1 |
| Initial to Final Layer Ratio | 41.0:1 | 112.9:1 | 65.1:1 | 20.5:1 |
| Decay Rate* | Moderate | Steep | Very Steep | Moderate |
| **Model-Specific Patterns** | | | | |
| Fisher Info per Parameter | Highest | Medium | Lowest | Medium |
| Layer Distribution | Front-loaded | Front-loaded | Front-loaded | Most balanced |
| Unique Feature | Layer 9-10 bump | Steady decline | Extreme decline | Final layer bump |

*Decay rate describes how quickly Fisher information diminishes across layers

Table 16: Value space analysis across Llama model variants

| Property | Llama-3.2-1B | Llama-3.2-3B | Llama-3.1-8B | Llama-3.1-8B-Instruct |
|---|---|---|---|---|
| *Value Space Metrics* | | | | |
| Relative Magnitude (Mean) | 0.6662 | 0.5415 | 0.5480 | 0.5384 |
| Directional Influence (Mean) | 0.6694 | 0.5421 | 0.5495 | 0.5400 |
| Directional Influence (Median) | 0.5926 | 0.5000 | 0.5000 | 0.5000 |
| Information Content (Mean) | 249.6769 | 410.9697 | 349.2795 | 373.5664 |
| Information Content (Median) | 282.5689 | 440.9005 | 373.0212 | 396.0780 |
| First-to-Last Layer Influence | -0.39 | -0.45 | -0.40 | -0.40 |
| *Attention-Value Correlation* | | | | |
| Attention Entropy (Mean) | 1.2627 | 1.0517 | 1.0058 | 1.0129 |
| Value Transformation Magnitude | 11.2118 | 9.7103 | 10.3658 | 10.1119 |
| First Layer Magnitude | 2.6703 | 2.4579 | 1.2628 | 1.2258 |
| Last Layer Magnitude | 106.1408 | 84.0693 | 133.1640 | 132.8909 |
| Geometric-Semantic Alignment | -0.3096 | -0.2921 | -0.3238 | -0.3599 |
| First Layer Alignment | -0.0318 | -0.0383 | -0.0839 | -0.0875 |
| Last Layer Alignment | -0.3128 | -0.2614 | -0.3456 | -0.3961 |
| Entropy-Magnitude Correlation | -0.0366 | 0.2273 | 0.3050 | 0.3229 |
| First-to-Last Layer Shift | -0.5148 to -0.5186 | -0.5349 to -0.5665 | -0.5002 to -0.5549 | -0.4974 to -0.5565 |

Table 17: Reference Frame Signatures Across Qwen Model Variants

| Property | Qwen2.5-3B | Qwen2.5-7B | Qwen2.5-7B-Instruct |
|---|---|---|---|
| Token specialization | "Ġthe" (36.6%) | "," (65.4%) | "," (64.4%) |
| $Betti_0$ (early) | 26.18 | 26.16 | 26.15 |
| $Betti_0$ (late) | 26.18 | 17.97 | 16.88 |
| $Betti_0$ change | 0.00 | -8.19 | -9.27 |
| Dim0 persistence (early) | 0.4183 | 0.2381 | 0.2382 |
| Dim0 persistence (late) | 0.2629 | 0.1543 | 0.1572 |
| Persistence change | -0.1555 | -0.0838 | -0.0810 |
| Max attention StdDev | 0.1274 | 0.1316 | 0.1296 |

In table 18 three Qwen models exhibit peak star-likeness in middle layers across multiple thresholds, with remarkably similar ratios ($1.00 - 1.05\times$ higher in middle layers). This consistency across

model scales (3B vs. 7B) and training objectives (base vs. instruct) suggests that the middle-layer organization is a fundamental characteristic of distributed reference frames.

Both 7B models show similar algebraic connectivity patterns, with peaks only at high thresholds (0.1-0.2). The 3B model also shows peaks at these thresholds but with slightly higher magnitude ($1.08 - 1.26\times$ vs. approximately $1.00\times$ in the 7B models). This suggests that while distributed reference frames consistently organize connectivity patterns differently than centralized frames, there are subtle scaling effects on the strength of these patterns.

Table 18: Spectral Signature of Distributed Reference Frame Signatures in Qwen Models

| Property | Qwen2.5-3B | Qwen2.5-7B | Qwen2.5-7B-Instruct |
|---|---|---|---|
| Position encoding | NTK-aware RoPE | NTK-aware RoPE | NTK-aware RoPE |
| Threshold effectiveness (0.1) | 0.64 | 0.25 | 0.32 |
| Algebraic connectivity (early) | 14.1862 | 16.2441 | 16.2151 |
| Algebraic connectivity (middle) | 12.3066 | 12.7960 | 12.9784 |
| Algebraic connectivity (late) | 11.5426 | 11.0412 | 11.2764 |
| Maximum connectivity | 20.1863 (layer 1) | 18.7700 (layer 1) | 18.4620 (layer 1) |
| Star-likeness (early) | 0.5349 | 0.5361 | 0.5360 |
| Star-likeness (middle) | 0.5349 | 0.5336 | 0.5337 |
| Star-likeness (late) | 0.5398 | 0.5041 | 0.5057 |
| Maximum star-likeness | 0.5655 (layer 33) | 0.5415 (layer 26) | 0.5431 (layer 26) |
| Degree centralization (early) | 0.5346 | 0.5237 | 0.5229 |
| Degree centralization (middle) | 0.5413 | 0.5372 | 0.5344 |
| Degree centralization (late) | 0.5528 | 0.4967 | 0.4932 |
| Maximum centralization | 0.6033 (layer 33) | 0.5625 (layer 26) | 0.5656 (layer 26) |
| Degree variance (early) | 119.9341 | 130.2675 | 126.5460 |
| Degree variance (middle) | 114.8901 | 117.3620 | 115.2299 |
| Degree variance (late) | 115.2607 | 104.4904 | 102.9542 |

The sign-flipping correlation pattern between Fiedler values and centralization metrics emerges as the most reliable signature of distributed reference frames. All Qwen models show negative correlations at low thresholds transitioning to positive correlations at higher thresholds, with the sign-flip consistently occurring around the 0.05 threshold. The 3B model shows slightly higher threshold sensitivity than the 7B models, remaining effective up to 0.1 rather than 0.05. This suggests that larger models with distributed reference frames may develop more specialized attention patterns that become disconnected at lower thresholds.

All three Qwen models show peak degree centralization in middle layers, but with interesting variations: the 3B model shows peaks primarily at high thresholds (0.05-0.2), while both 7B models show peaks across multiple thresholds. This suggests that scale may influence how distributed reference frames organize their centralization patterns, with larger models developing more consistent middle-layer centralization.

The table 20 shows distinctive attention sink patterns across the Qwen model family, providing quantitative evidence for distributed reference frames. All models demonstrate a three-phase pattern with positive KL reduction in early layers (+0.078 to +0.109), stronger negative reduction in middle layers (-0.026 to -0.051), and moderated negative values in late layers (-0.008 to -0.018). This indicates that reference points serve different functions at different network depths, initially establishing geometric stability, then actively shaping information geometry in middle layers, before stabilizing in deeper layers.

Maximum sink concentration occurs in deep layers (23-25) rather than early layers, reaching 84-92% - a key signature of distributed reference frames. The 3B model shows stronger contrasts between layer regions, particularly with more negative KL reduction in middle layers (-0.0514 vs -0.0266/-0.0299), suggesting smaller models may rely more heavily on distributed reference structures for computational efficiency.

The layer with minimum KL reduction (most negative impact when removing sinks) is consistently layer 5 across all models, while maximum concentration appears deeper in the network. This

Table 19: Key Fiedler Value Correlations across Qwen models at different thresholds

| Threshold | Property | Mean Correlation | Pattern |
|---|---|---|---|
| 0.001 | Centralization vs. Fiedler | -0.4383 | Moderate negative |
| 0.02 | Centralization vs. Fiedler | -0.6152 | Strong negative |
| 0.05 | Centralization vs. Fiedler | -0.0482 | Near zero (transition point) |
| 0.1 | Centralization vs. Fiedler | 0.5565 | Sign flip to positive |
| 0.001 | Density vs. Fiedler | -0.0996 | Slight negative |
| 0.02 | Density vs. Fiedler | 0.6835 | Strong positive |
| 0.1 | Density vs. Fiedler | 0.1147 | Weak positive |

separation between maximum effect and maximum concentration further supports the multi-pointed manifold structure hypothesized for distributed reference frames, where coordination is achieved through multiple specialized reference points rather than a single dominant one.

Table 20: Attention Sink Analysis Across Qwen Model Variants

| Property | Qwen2.5-3B | Qwen2.5-7B | Qwen2.5-7B-Instruct |
|---|---|---|---|
| KL Reduction (0-3 layers) | +0.1090 | +0.0859 | +0.0780 |
| KL Reduction (5-15 layers) | -0.0514 | -0.0266 | -0.0299 |
| KL Reduction (17-35 layers) | -0.0084 | -0.0177 | -0.0134 |
| Maximum KL reduction | 0.1329 (Layer 0) | 0.1136 (Layer 1) | 0.1045 (Layer 1) |
| Minimum KL reduction | -0.3058 (Layer 5) | -0.1629 (Layer 5) | -0.1515 (Layer 5) |
| Sink concentration (0-3 layers) | 39.59% | 48.58% | 48.13% |
| Sink concentration (5-15 layers) | 78.03% | 77.88% | 76.93% |
| Sink concentration (17-35 layers) | 77.60% | 74.42% | 72.95% |
| Maximum sink concentration | 91.89% (Layer 25) | 85.75% (Layer 23) | 84.27% (Layer 23) |
| Optimal threshold value | 0.8 | 0.8 | 0.8 |
| Number of analyzed layers | 19 | 15 | 15 |
| Sequence length | 128 | 128 | 128 |

Table 21: Fisher Information Distribution in Qwen Models

| Architectural Pattern | Qwen2.5-3B | Qwen2.5-7B | Qwen2.5-7B-I |
|---|---|---|---|
| **Component Importance (Percentage of Total Fisher Norm)** | | | |
| Attention Mechanism | 10.4% | 29.7% | 30.2% |
| MLP Components | 84.7% | 51.8% | 56.1% |
| Embedding | 4.9% | 2.1% | 1.7% |
| **Layer Distribution Patterns** | | | |
| Key Processing Layers | Layer 2 (53.7%) | Layers 0-6 (54.9%) | Layers 0-6 (56.6%) |
| Early Layers (0-9) | 78.7% | 64.7% | 63.9% |
| Middle Layers (10-27) | 11.0% | 14.9% | 17.1% |
| Final Layers | 7.2% (28-35) | 3.9% (26-27) | 3.9% (26-27) |
| **Layer Decay Characteristics** | | | |
| Peak Layer Value | Layer 2: 136,593 | Layer 0: 18,791 | Layer 1: 18,359 |
| Peak to Minimum Ratio | 432:1 | 26:1 | 18.6:1 |
| Decay Rate* | Very steep after peak | Steady, gradual | Steady, moderate |
| **Model-Specific Patterns** | | | |
| Fisher Info per Parameter | Medium | Lowest | Low |
| Layer Distribution | Single peak & steep drop | Heavy & gradual decline | Heavy & gradual decline |
| Unique Feature | Layer 30 bump (14,067) | Final layers bump | Smoother distribution |

*Decay rate describes how quickly Fisher information diminishes across layers

Table 22: Value Space analysis across Qwen model variants

| Property | Qwen2.5-3B | Qwen2.5-7B | Qwen2.5-7B-Instruct |
|---|---|---|---|
| *Value Space Metrics* | | | |
| Relative Magnitude (Mean) | 0.4655 | 0.4890 | 0.4897 |
| Directional Influence (Mean) | 0.5313 | 0.5521 | 0.5525 |
| Directional Influence (Median) | 0.5000 | 0.5000 | 0.5000 |
| Information Content (Mean) | 1792.7866 | 7161.5074 | 7162.1374 |
| Information Content (Median) | 2054.0861 | 8431.1512 | 8519.9060 |
| First-to-Last Layer Influence | -0.15 | +0.18 | +0.18 |
| *Attention-Value Correlation* | | | |
| Attention Entropy (Mean) | 1.5390 | 1.4449 | 1.4711 |
| Value Transformation Magnitude | 31.6244 | 16.2740 | 14.9721 |
| First Layer Magnitude | 21.4661 | 15.0259 | 14.7112 |
| Last Layer Magnitude | 358.9528 | 222.3867 | 191.2135 |
| Geometric-Semantic Alignment | -0.1472 | -0.0426 | -0.0395 |
| First Layer Alignment | 0.0676 | 0.0569 | 0.0578 |
| Last Layer Alignment | -0.0349 | 0.0000 | 0.0000 |
| Entropy-Magnitude Correlation | -0.0842 | -0.2325 | -0.2329 |
| First-to-Last Layer Shift | -0.4763 to -0.2981 | -0.7045 to -0.0459 | -0.6958 to -0.0501 |

# G  Gemma and Mistral

As we can see from table 23, both models show 100% specialization on their respective beginning-of-sequence tokens ( in Mistral, <bos> in Gemma). This perfect specialization is the hallmark of centralized reference frames, establishing a single dominant reference point that serves as the universal origin for the representation manifold. Both models maintain identical $Betti_0$ counts from early to late layers ($27.24 \rightarrow 27.24$ in Mistral, $26.11 \rightarrow 26.11$ in Gemma), indicating complete topological stability of component structure. This stability is characteristic of centralized frames, where the component organization remains fixed throughout the network. Both models show monotonically increasing persistence values through layers, with Mistral showing a +0.0796 increase and Gemma showing a remarkable +0.3163 increase. This strengthening of reference point significance through network depth is a defining property of centralized frames that differentiates them from distributed frames (which show decreasing persistence) and hybrid frames like Pythia. Like other decoder models, both Mistral and Gemma show no loops ($Betti_1 = 0.00 = 0.00$) at any layer, contrasting sharply with the complex cyclic structures found in bidirectional encoder models like BERT and RoBERTa.

Table 23: Centralized Reference Frame Signatures in Mistral and Gemma Models

| Property | Mistral-7B-v0.3 | Gemma-7B |
|---|---|---|
| Position encoding | RoPE | RoPE (Modified) |
| Token specialization | 100% on  | 100% on <bos> |
| $Betti_0$ (early) | 27.24 | 26.11 |
| $Betti_0$ (late) | 27.24 | 26.11 |
| $Betti_0$ change | 0.00 | 0.00 |
| $Betti_1$ (early) | 0.00 | 0.00 |
| $Betti_1$ (late) | 0.00 | 0.00 |
| $Betti_1$ change | 0.00 | 0.00 |
| Dim0 persistence (early) | 0.0310 | 0.2015 |
| Dim0 persistence (late) | 0.1105 | 0.5179 |
| Persistence change | +0.0796 | +0.3163 |
| Max attention StdDev | 0.1503 (layer 24) | 0.1404 (layer 21) |

Despite sharing the same fundamental reference frame type, Mistral and Gemma show important differences in how they implement their centralized frames: for example, gemma starts with much higher persistence values (0.2015) compared to Mistral (0.0310), indicating a stronger initial reference point structure. This suggests Gemma establishes a more dominant centralized reference from the earliest layers. The most striking difference is in the magnitude of persistence growth. Gemma's persistence increase (+0.3163) is nearly four times larger than Mistral's (+0.0796), resulting in an extremely high final persistence value of 0.5179. This suggests Gemma's centralized reference becomes exceptionally dominant in deeper layers. While both models show peak attention standard deviation in similar layers (24 for Mistral, 21 for Gemma), the overall attention distributions and specialization patterns across layers likely differ in subtle ways not fully captured in the topological metrics.

In table 24 we can see the algebraic connectivity (Fiedler values) across both models shows a distinctive pattern: for Mistral-7B, we observe a clear "inverted U" pattern where connectivity peaks in middle layers (13.3343) compared to early (13.0303) and late layers (12.0898). This pattern suggests that middle layers serve as a critical transition point in the reference frame structure—they balance information flow between the initial reference frame establishment and the subsequent semantic processing. In contrast, Gemma-7B shows a monotonic decrease in connectivity from early (13.0545) to middle (11.0557) to late layers (9.9455). This different pattern suggests that Gemma implements a distinctly different reference frame strategy, potentially establishing stronger reference points earlier in the network.

Table 24: Spectral Graph Analysis of Reference Frame Structures

| Property | Mistral-7B-v0.3 | Gemma-7B |
|---|---|---|
| Most effective threshold | 0.001 | 0.001 |
| *Algebraic Connectivity (Fiedler Values)* | | |
| Early layers (avg) | 13.0303 | 13.0545 |
| Middle layers (avg) | 13.3343 | 11.0557 |
| Late layers (avg) | 12.0898 | 9.9455 |
| Maximum value | 20.7132 (layer 0) | 15.1970 (layer 1) |
| *Star-likeness* | | |
| Early layers (avg) | 0.5371 | 0.5354 |
| Middle layers (avg) | 0.5345 | 0.5344 |
| Late layers (avg) | 0.5372 | 0.5344 |
| Maximum value | 0.5441 (layer 2) | 0.5382 (layer 14) |
| *Degree Centralization* | | |
| Early layers (avg) | 0.5475 | 0.5357 |
| Middle layers (avg) | 0.5411 | 0.5488 |
| Late layers (avg) | 0.5564 | 0.5650 |
| Maximum value | 0.5942 (layer 2) | 0.5948 (layer 22) |

One of the most striking findings in the Mistral data is the strong negative correlation (-0.9405) between algebraic connectivity and degree centralization. This trade-off represents a fundamental tension in transformer architecture design, between establishing stable reference points (high centralization) and enabling efficient information flow (high connectivity). The fact that this correlation is so strong (-0.9405) suggests this is not an incidental pattern but a core organizing principle. Both models show remarkably similar star-likeness metrics at the baseline threshold (0.001), because both models converge on very similar star-like structures despite their different architectural designs. The fact that star-likeness metrics remain consistently high (>0.53) across all layers in both models indicates that reference points remain essential throughout the entire network depth, not just in early stages.

The strong positive correlation (0.9481) between Fiedler values and degree variance in Mistral suggests that as the model balances compression and relevance (as measured by connectivity), it simultaneously increases the variance in how information is distributed, creating specialized processing structure.

In table 25 we can see the predominantly negative KL reduction values throughout middle layers (-0.0557 for Mistral, -0.0590 for Gemma) indicate that removing attention sinks increases the KL

Table 25: Attention Sink KL Divergence Analysis for Reference Frame Models

| Property | Mistral-7B-v0.1 | Gemma-7B |
|---|---|---|
| *KL Reduction (t=0.8)* | | |
| Early layers (0-5) | -0.0576 | 0.0105 |
| Middle layers (7-19) | -0.0557 | -0.0590 |
| Late layers (21-31) | -0.0703 | -0.0330 |
| First layer | 0.1342 | 0.1435 |
| Final layer | 0.0111 | 0.0576 |
| *Sink Concentration (t=0.8)* | | |
| Early layers (0-5) | 89.62% | 73.59% |
| Middle layers (7-19) | 81.55% | 80.28% |
| Late layers (21-31) | 84.90% | 80.75% |
| First layer | 86.95% | 61.46% |
| Final layer | 63.64% | 53.51% |
| *Layer-specific Patterns* | | |
| Highest sink concentration | 95.44% (layer 21) | 91.57% (layer 21) |
| Lowest sink concentration | 63.64% (layer 31) | 53.51% (layer 27) |
| Most negative KL reduction | -0.1907 (layer 3) | -0.1218 (layer 11) |
| *Threshold Effects (final layer)* | | |
| KL reduction (t=0.8) | 0.0111 | 0.0576 |
| KL reduction (t=0.9) | -0.0257 | 0.0391 |
| KL reduction (t=0.95) | -0.0413 | 0.0285 |
| *Threshold Effects (sink concentration)* | | |
| t=0.8 to t=0.95 change | -23.9% points | -18.8% points |

divergence between attention distributions. What's particularly fascinating is that both models show positive KL reduction values in their first layer (0.1342 for Mistral, 0.1435 for Gemma) and final layer (0.0111 for Mistral, 0.0576 for Gemma). This U-shaped pattern suggests that reference frames serve different information-geometric functions at different depths in the network

Table 26: Key Fiedler Value Correlations in Mistral-7B and Gemma-7B Models

| Threshold | Property | Mistral-7B | Gemma-7B |
|---|---|---|---|
| 0.01 | Centralization vs. Fiedler | -0.7787 | -0.7429 |
| 0.05 | Centralization vs. Fiedler | -0.1273 | -0.3591 |
| 0.10 | Centralization vs. Fiedler | 0.5893 | 0.2162 |
| 0.01 | Density vs. Fiedler | 0.7345 | 0.6935 |
| 0.10 | Density vs. Fiedler | 0.5764 | 0.4387 |

Mistral shows extremely high sink concentration in early layers (89.62% at t=0.8), which then decreases in middle layers (81.55%) before slightly increasing again in late layers (84.90%). This pattern aligns with your description of centralized reference frames where a single dominant reference point establishes a universal origin. Gemma shows a markedly different pattern with significantly lower sink concentration in early layers (73.59% at t=0.8), which then increases in middle layers (80.28%) and remains stable in late layers (80.75%). This suggests a more distributed reference frame that develops progressively through the network depth.

One of the most striking differences between the models is in their first-layer sink concentration, the difference (25.49 percentage points) likely reflects the different position encoding implementations you described in your theoretical framework. Mistral's standard RoPE implementation creates a stronger positional bias toward the first token, facilitating a more centralized reference frame. Gemma's modified position encoding appears to reduce this positional bias, enabling a more distributed reference frame with multiple weaker reference points.

The fact that both models ultimately achieve similarly high maximum sink concentrations (95.44% vs. 91.57%, both at layer 21) suggests that strong reference points are mathematically necessary despite these architectural differences.

Table 27: Fisher Information Distribution in Mistral and Gemma Models

| Architectural Pattern | Mistral-7B-v0.1 | Gemma-7B |
|---|---|---|
| **Component Importance (Percentage of Total Fisher Norm)** | | |
| Attention Mechanism | 58.8% | High* |
| MLP Components | 31.8% | Very High* |
| Embedding | 8.7% | Low* |
| **Layer Distribution Patterns** | | |
| Key Processing Layers | Layer 1 (44.0%) | Layers 1, 26-27 (77.2%) |
| Early Layers (0-9) | 73.9% | High* |
| Middle Layers (10-19) | 11.6% | Medium* |
| Final Layers (20-31) | 4.4% | Very High* |
| **Layer Decay Characteristics** | | |
| Peak Layer Value | Layer 1: 3,053,101 | Layer 1: 13,804,803 |
| Peak to Minimum Ratio | 165:1 | High* |
| Decay Pattern | Sharp drop, slow decline | U-shaped distribution |
| **Model-Specific Patterns** | | |
| Fisher Info Distribution | Front-loaded, with minor bump at end | Strong bi-modal distribution |
| Layer Distribution | Steep initial drop, then plateau | Sharp drop after Layer 1, rise in final layers |
| Unique Feature | Small bumps at layers 12 and 18 | Extreme values in layers 0 and 27 |

*Exact values affected by overflow in Gemma computation (Infinity reported in some components)

Table 28: Value space analysis for Gemma and Mistral models

| Property | google/gemma-7b | mistralai/Mistral-7B-v0.1 |
|---|---|---|
| *Value Space Metrics* | | |
| Relative Magnitude (Mean) | 0.5796 | 0.9758 |
| Directional Influence (Mean) | 0.5830 | 0.9771 |
| Directional Influence (Median) | 0.5312 | 0.9827 |
| Information Content (Mean) | 307.7589 | 179.1840 |
| Information Content (Median) | 309.7665 | 189.7641 |
| First-to-Last Layer Influence | +0.33 | +0.01 |
| *Attention-Value Correlation* | | |
| Attention Entropy (Mean) | 1.2289 | 1.1524 |
| Value Transformation Magnitude | 28.2835 | 14.4851 |
| First Layer Magnitude | 205.1822 | 0.3779 |
| Last Layer Magnitude | 446.5816 | 355.8180 |
| Geometric-Semantic Alignment | -0.4012 | -0.1386 |
| First Layer Alignment | -0.6124 | -0.1006 |
| Last Layer Alignment | -0.3177 | -0.3320 |
| Entropy-Magnitude Correlation | -0.2408 | 0.2264 |
| First-to-Last Layer Shift | -0.3443 to -0.1462 | -0.4777 to -0.1529 |

# H  Encoder only (BERT and Roberta)

In table 29, we can see that both models exhibit a striking pattern of dual specialization where attention heads focus on different special tokens depending on layer depth: early layers show perfect (100%) specialization on beginning tokens ([CLS] in BERT,  in RoBERTa); and middle/late layers show perfect (100%) specialization on ending tokens ([SEP] in BERT,  in RoBERTa). This layer-specific specialization pattern creates a "bipolar" reference structure that differs fundamentally from both the single-point centralization in Llama and the distributed multi-point structure in Qwen.

Both encoder models exhibit remarkably high initial topological complexity: high $Betti_1$ values (15.40 in BERT, 19.69 in RoBERTa) indicate numerous loops/cycles in early layers, and this contrasts sharply with decoder models (Llama, Qwen) which show virtually no loops ($Betti_1 = 0.00$). This high initial complexity reflects how bidirectional attention creates rich interconnected structures that allow tokens to reference each other in complex ways rather than primarily referencing a central point or distributed landmarks.

Table 29: Bidirectional Reference Frame Signatures in Encoder Models

| Property | BERT-base-uncased | XLM-RoBERTa-large |
|---|---|---|
| Token specialization | Early: 100% on [CLS]
Middle: 100% on [SEP] | Early: 100% on \<s\>
Late: 100% on \</s\> |
| $Betti_0$ (early) | 4.48 | 22.68 |
| $Betti_0$ (late) | 4.33 | 1.69 |
| $Betti_0$ change | -0.14 | -20.99 |
| $Betti_1$ (early) | 15.40 | 19.69 |
| $Betti_1$ (late) | 2.79 | 3.11 |
| $Betti_1$ change | -12.61 | -16.58 |
| Dim0 persistence (early) | 0.0453 | 0.0521 |
| Dim0 persistence (late) | 0.0194 | 0.0156 |
| Persistence change | -0.0259 | -0.0365 |
| Max attention pattern | Layer 0: 0.3362
Layer 6: 0.8765
Layer 12: 0.8135 | Layer 0: 0.1731
Layer 11: 0.4731
Layer 23: 0.3540 |

As information flows through the network, bidirectional reference frames undergo dramatic topological simplification, for example, we can see substantial reduction in loops ($Betti_1$ decreases by 12.61 in BERT, 16.58 in RoBERTa) Component integration in RoBERTa is particularly interesting, since $Betti_1$ decreases from 22.68 to 1.69. This simplification pattern suggests that bidirectional frames start with complex relationships that gradually consolidate around key reference points as information is processed.

Both models show decreasing persistence values from early to late layers, similar to distributed frames but unlike centralized frames:

– BERT: $0.0453 \rightarrow 0.0194(-0.0259)$

– RoBERTa: $0.0521 \rightarrow 0.0156(-0.0365)$

This suggests that feature significance weakens through layers as the model transitions from complex initial representations to more specialized final representations.

In table 30 we can see that while decoder models (Llama, Mistral, Gemma) show their maximum connectivity in layer 0, these bidirectional models show maximum connectivity in early but not initial layers (layer 2 for BERT, layer 1 for XLM-RoBERTa). This subtle shift indicates a fundamentally different geometric organization strategy. What's more remarkable is the dramatic drop in connectivity from early to middle layers ($46.69 \rightarrow 23.79$ in BERT, $38.27 \rightarrow 31.25$ in XLM-RoBERTa), which contrasts sharply with the gentle declines or even increases seen in decoder models. This suggests a rapid geometric reorganization as information moves through the network.

The U-shaped algebraic connectivity pattern in BERT (high→low→medium) differs completely from both centralized reference frames (high→high→medium) and distributed reference frames (high→medium→low). This suggests a fundamentally different geometric organization. Both models show much lower baseline star-likeness values (0.33-0.39) compared to decoder models (0.53-0.54), suggesting less reliance on single reference points. This aligns with your theory that bidirectional models establish dual reference points rather than a single dominant one. Unlike decoder models, these architectures show peak degree centralization in middle layers at the low threshold (0.1579 for BERT, 0.1274 for XLM-RoBERTa), indicating that the reference structure evolves substantially through network depth. The effectiveness scores drop sharply at the 0.1 threshold (0.42 for BERT,

Table 30: Bidirectional Reference Frame Signatures in Encoder Models

| Property | BERT-base-uncased | XLM-RoBERTa-large |
|---|---|---|
| Position encoding | Absolute | Absolute |
| Most effective threshold | 0.01 | 0.01 |
| Threshold effectiveness (0.1) | 0.42 | 0.38 |
| Algebraic connectivity (early) | 46.6913 | 38.2726 |
| Algebraic connectivity (middle) | 23.7880 | 31.2481 |
| Algebraic connectivity (late) | 27.3752 | 22.5304 |
| Maximum connectivity | 52.4496 (layer 2) | 55.3003 (layer 1) |
| Star-likeness (early) | 0.3274 | 0.3491 |
| Star-likeness (middle) | 0.3680 | 0.3550 |
| Star-likeness (late) | 0.3585 | 0.3941 |
| Maximum star-likeness | 0.4121 (layer 7) | 0.4383 (layer 20) |
| Degree centralization (early) | 0.0467 | 0.0786 |
| Degree centralization (middle) | 0.1579 | 0.1274 |
| Degree centralization (late) | 0.1208 | 0.2234 |
| Maximum centralization | 0.2730 (layer 7) | 0.3556 (layer 19) |
| Degree variance (early) | 28.7433 | 98.2550 |
| Degree variance (middle) | 50.4899 | 50.8467 |
| Degree variance (late) | 52.4320 | 89.7002 |

0.38 for XLM-RoBERTa) compared to decoder models, indicating more complex attention patterns that can't be simplified to binary relationships.

In table 32 we can see that the most insteresting pattern in the data is the distinctive U-shaped KL reduction profile across network depth. Both models show positive KL reduction in their first layer (0.0392 for BERT, 0.0635 for XLM-RoBERTa) and final or near-final layers (XLM-RoBERTa shows 0.0774 in layer 23). The larger model, roberta, shows a more dramatic U-shaped pattern, with stronger positive KL reduction in the final layer (0.0774 vs. -0.0179). This suggests that increased model capacity allows for more distinct reference points at sequence boundaries. Also, Roberta shows its peak negative KL reduction in a deeper layer (layer 9 vs. layer 5 in BERT), indicating that larger models can maintain the initial coordinate system longer before transitioning to the integration phase.

Table 31: Key Fiedler Value Correlations in Encoder Models with Bidirectional Reference Frames

| Threshold | Property | BERT-base | XLM-RoBERTa |
|---|---|---|---|
| 0.001 | Centralization vs. Fiedler | 0.1801 | 0.3242 |
| 0.02 | Centralization vs. Fiedler | -0.8442 | -0.6672 |
| 0.1 | Centralization vs. Fiedler | 0.3233 | -0.3302 |
| 0.2 | Centralization vs. Fiedler | 0.2859 | 0.2347 |
| 0.01 | Density vs. Fiedler | 0.8227 | 0.6146 |

Both models show substantial degradation in KL reduction at higher thresholds in early layers (-45.2% for BERT, -58.1% for XLM-RoBERTa from t=0.8 to t=0.95). This suggests that the initial reference point relies on a broad distribution of attention weights. Both bidirectional models show remarkably low sink concentration in their first layers (32.07% for BERT, 28.19% for XLM-RoBERTa), much lower than the 80-90% concentration typically seen in decoder models. This indicates a fundamentally different approach to establishing initial coordinate systems.

The clear differences between bidirectional and decoder models confirm that architectural choices fundamentally shape the geometric organization of the representation space. The use of absolute position embeddings in these models corresponds to their bipolar reference structure, supporting your claim about position encoding implementations influencing reference frame types. And the U-shaped KL reduction pattern provides perhaps the clearest evidence yet for the characterization

Table 32: KL Divergence Analysis of Bidirectional Reference Frames

| Property | BERT-base-uncased | XLM-RoBERTa-large |
|---|---|---|
| *KL Reduction (t=0.8)* | | |
| Early layers (0-3) | -0.0243 | -0.0306 |
| Middle layers (5-9) | -0.1121 | -0.2156 |
| Late layers (11+) | -0.0179 | -0.0826 |
| First layer | 0.0392 | 0.0635 |
| Final layer | -0.0179 | 0.0774 |
| Maximum negative | -0.1733 (layer 5) | -0.2574 (layer 9) |
| *Sink Concentration (t=0.8)* | | |
| Early layers (0-3) | 51.10% | 51.06% |
| Middle layers (5-9) | 71.73% | 75.06% |
| Late layers (11+) | 77.20% | 73.11% |
| First layer | 32.07% | 28.19% |
| Final layer | 77.20% | 44.39% |
| Maximum concentration | 77.20% (layer 11) | 85.22% (layer 21) |
| *Layer-specific Patterns* | | |
| KL U-shape (first→final) | Yes (+) | Yes (+) |
| Mid-layer KL reduction peak | Layer 5 | Layer 9 |
| Late layer low concentration | No | Yes (44.39%) |
| *Threshold Effects* | | |
| t=0.8 to t=0.95 deepening | -137.9% | -15.5% |
| Early KL reduction change | -45.2% | -58.1% |
| Middle KL reduction change | -22.2% | +15.5% |
| Late KL reduction change | -117.9% | -2.1% |

of bidirectional models as establishing a bipolar manifold with reference points at both sequence boundaries.

Table 33: Architectural Pattern Analysis Based on Fisher Information Distribution in Encoder Models

| Architectural Pattern | BERT-base-uncased | XLM-RoBERTa-large |
|---|---|---|
| **Component Importance (Percentage of Total Fisher Norm)** | | |
| Attention Mechanism | 22.5% | 71.7% |
| MLP Components | 44.5% | 27.8% |
| Embedding | 31.3% | 0.2% |
| **Layer Distribution Patterns** | | |
| Key Processing Layers | Layers 0-1, 11 (32.7%) | Layers 7-10 (64.7%) |
| Early Layers (0-3) | 35.9% | 10.5% |
| Middle Layers (4-8) | 16.1% | 52.4% |
| Final Layers (9+) | 13.0% | 36.9% |
| **Layer Decay Characteristics** | | |
| Peak Layer Value | Layer 1: 666 | Layer 9: 2,504,416 |
| Peak to Minimum Ratio | 5.9:1 | 302.6:1 |
| Decay Pattern | Gradual decline with end spike | Inverted U-shape with end spike |
| **Model-Specific Patterns** | | |
| Fisher Info Distribution | Relatively balanced | Highly concentrated in middle |
| Layer Distribution | Early and final emphasis | Middle-heavy with final spike |
| Unique Feature | High embedding importance | Dramatic middle-layer concentration |

Table 34: Value space analysis for BERT and XLM-RoBERTa models

| Property | BERT-base-uncased | XLM-RoBERTa-large |
|---|---|---|
| *Value Space Metrics* | | |
| Relative Magnitude (Mean) | 0.5289 | 0.5928 |
| Directional Influence (Mean) | 0.8141 | 0.9394 |
| Directional Influence (Median) | 0.8239 | 0.9683 |
| Information Content (Mean) | 4.0998 | 13.7629 |
| Information Content (Median) | 4.3081 | 14.3960 |
| First-to-Last Layer Influence | +0.19 | +0.20 |
| *Attention-Value Correlation* | | |
| Attention Entropy (Mean) | 2.3200 | 2.1957 |
| Value Transformation Magnitude | 8.6058 | 8.1538 |
| First Layer Magnitude | 9.4719 | 19.7373 |
| Last Layer Magnitude | 11.5637 | 23.8058 |
| Geometric-Semantic Alignment | -0.1200 | 0.0013 |
| First Layer Alignment | 0.1246 | 0.2065 |
| Last Layer Alignment | -0.3608 | -0.3982 |
| Entropy-Magnitude Correlation | 0.2940 | 0.2326 |
| First-to-Last Layer Shift | -0.1935 to -0.1043 | -0.1552 to 0.6409 |

# I   Phi

The results from microsoft/phi-2 shown in table 35 reveal a clear distributed reference frame architecture characterized by distinctive patterns across multiple analytical methodologies. The Fisher information distribution demonstrates a balanced allocation between attention mechanisms (37.4%) and MLP components (32.7%), contrasting with the attention-dominated profiles typically observed in centralized reference frame models. Rather than concentrating information processing in early layers, phi-2 exhibits a distinctive triple-peaked distribution with significant Fisher norm values in layers 0-1 (5.5%), a secondary peak around layer 26 (262.81), and pronounced concentration in final layers 29-31 (49.6% of total Fisher norm). This multi-peaked pattern represents a fundamental departure from the early-layer concentrated processing found in centralized reference frame architectures like LLaMA. The KL divergence analysis provides particularly compelling evidence for the distributed reference frame classification through its characteristic three-phase pattern. Early layers (particularly layer 0) show positive KL reduction values (+0.1416 at threshold 0.8), indicating that removing attention sinks actually improves information flow at these stages. This transitions sharply to negative values in middle layers (ranging from -0.1525 to -0.1995), followed by stronger negative values in deeper layers (peaking at -0.2787 in layer 25). This progression reflects a fundamentally different approach to utilizing reference points compared to centralized models, which typically show consistently negative KL reduction across all network depths. The attention sink concentration metrics further reinforce this pattern, showing a progressive buildup from relatively low values in early layers (35.23% in layer 0) to very high concentration in deep layers (97.19% in layer 25), rather than the consistently high concentration characteristic of centralized reference frames. Spectral graph analysis identifies the mathematical signature that definitively marks phi-2 as implementing a distributed reference frame. The model exhibits a characteristic sign-flipping correlation pattern between Fiedler values (algebraic connectivity) and centralization metrics, shifting from negative correlation at low thresholds (-0.4518 at 0.001) to strong positive correlation at higher thresholds (+0.6009 at 0.1). This pattern closely mirrors the signature observed in Qwen models (-0.46 $\rightarrow$ +0.61) that use similar NTK-aware scaled rotary position embeddings. Layer-specific correlations reveal particularly strong negative correlations in early layers (-0.8796 in layer 0) at low thresholds, with progressive transition to positive correlations at higher thresholds through network depth. Topological analysis supports these findings, showing substantial evolution in attention structure across network depth. The $Betti_0$ numbers (connected components) increase dramatically from 0.0 in layer 0 to 25.99 in layer 31, indicating progressive fragmentation of attention rather than the stable topolog-

Table 35: Reference Frame Analysis of Microsoft/Phi-2

| Analysis Category | Microsoft/Phi-2 |
|---|---|
| **Fisher Information Distribution** | |
| Attention Mechanism | 37.4% (3905.88) |
| MLP Components | 32.7% (3419.73) |
| Embedding | 1.0% (105.28) |
| Other Components | 28.9% (3012.81) |
| **Layer Distribution Patterns** | |
| Key Processing Layers | Layers 0-1 (5.5%), 29-31 (49.6%) |
| Early Layers (0-10) | 19.8% (2066.12) |
| Middle Layers (11-28) | 17.4% (1818.65) |
| Final Layers (29-31) | 49.6% (5181.61) |
| **Layer Decay Characteristics** | |
| Peak Layer Value | Layer 30: 2973.23 |
| Secondary Peaks | Layer 0: 342.85, Layer 26: 262.81 |
| Peak to Minimum Ratio | 30.3:1 (Layer 30 vs Layer 17) |
| Multi-peaked Pattern | Yes (Early, Middle, Late) |
| **KL Divergence Analysis** | |
| Early Layer KL Red. (t=0.8) | Positive (Layer 0: +0.1416) |
| Middle Layer KL Red. (t=0.8) | Negative (Layer 11: -0.1525) |
| Deep Layer KL Red. (t=0.8) | Stronger Negative (Layer 25: -0.2787) |
| KL Pattern | Three-phase (Positive $\rightarrow$ Negative $\rightarrow$ Stronger Negative) |
| **Attention Sink Concentration** | |
| Early Layer Concentration | Low (Layer 0: 35.23%) |
| Middle Layer Concentration | High (Layer 11: 92.37%) |
| Deep Layer Concentration | Very High (Layer 25: 97.19%) |
| Concentration Growth | Progressive (35.23% $\rightarrow$ 97.19%) |
| **Spectral Graph Analysis** | |
| Low Threshold Correlation (0.001) | Negative (Fiedler vs. Centralization: -0.4518) |
| High Threshold Correlation (0.1) | Positive (Fiedler vs. Centralization: +0.6009) |
| Correlation Pattern | Sign-flipping (-0.4518 $\rightarrow$ +0.6009) |
| Signature Feature | Early layer strong negative correlation (-0.8796 in layer 0) |
| **Topological Features** | |
| $Betti_0$ Progression | 0.0 $\rightarrow$ 25.99 (Layers 0 $\rightarrow$ 31) |
| $Betti_1$ Values | Consistently 0.00 (No loops) |
| Connected Component Change | +25.99 (significant fragmentation) |
| Structural Evolution | Progressive disconnection across depth |
| **Reference Frame Classification** | |
| Frame Type | Distributed Reference Frame |
| Position Encoding | NTK-aware scaled RoPE |
| Key Evidence | Three-phase KL pattern, Sign-flipping correlation, |
| | Multi-peaked Fisher information, Progressive sink concentration |

ical structure maintained in centralized reference frame models. This fragmentation reflects how distributed reference frames implement a more flexible coordinate system that can adapt to different computational needs across network depth.

# J    Pythia Family

Examining the Pythia family of models (Table 37) reveals how centralized reference frames evolve with increasing model scale. While Pythia models exhibit the defining characteristics of centralized reference frames, we observe a systematic scaling relationship between model size and reference

Table 36: Value space analysis for Microsoft Phi-2

| Property | microsoft/phi-2 |
|---|---|
| *Value Space Metrics* | |
| Relative Magnitude (Mean) | 0.5041 |
| Directional Influence (Mean) | 0.5334 |
| Directional Influence (Median) | 0.5000 |
| Information Content (Mean) | 593.5976 |
| Information Content (Median) | 710.1380 |
| First-to-Last Layer Influence | -0.23 |
| *Attention-Value Correlation* | |
| Attention Entropy (Mean) | 0.9237 |
| Value Transformation Magnitude | 23.9090 |
| First Layer Magnitude | 22.6574 |
| Last Layer Magnitude | 0.0000 |
| Geometric-Semantic Alignment | -0.1478 |
| First Layer Alignment | 0.0370 |
| Last Layer Alignment | 0.0000 |
| Entropy-Magnitude Correlation | 0.1305 |
| First-to-Last Layer Shift | -0.5548 to 0.0000 |

point strength. As model scale increases from 2.8B to 12B parameters, token specialization on the most attended token (consistently "Ġthe") increases proportionally from 31.0% to 42.2%. The number of specialized attention heads also scales with model size, increasing from 1 to 4 across the family. This suggests that larger models develop more pronounced centralized reference structures, potentially enabling more efficient information routing through the network. Topologically, Pythia models maintain remarkably stable $Betti_0$ numbers ( 25.7) across all model scales and through network depth, confirming the centralized reference frame pattern. However, we observe a consistent decrease in persistence values from early to late layers, with larger models showing more dramatic reductions in persistence (-0.1065 in 2.8B to -0.1671 in 12B). This counter-intuitive finding suggests that while larger models establish stronger reference points, they simultaneously develop more nuanced relationships between these reference points and contextual tokens.

The maximum attention standard deviation shifts to earlier layers as model scale increases (from layer 16 in 2.8B to layer 9 in 12B), indicating that larger models establish their reference structures more efficiently and earlier in the processing pipeline. This aligns with our temporal emergence findings that reference frames develop more rapidly during training in larger models.

In table 37 we can see that as model size increases from 2.8B to 12B parameters, there is a clear strengthening of centralized reference frame structures. For example, the consistency of attention to the token "Ġthe" increases systematically with model scale ($31.0\% \rightarrow 36.8\% \rightarrow 42.2\%$). This suggests that larger models develop more robust reference points, supporting the theory that reference frames are fundamental geometric adaptations. Then, we can see that the number of specialized heads increases from just 1 in the smallest model to 4 in the largest. This indicates that with greater capacity, models allocate more resources to establishing reference frames, underscoring their importance.

The Betti numbers show consistency in the basic topological structure: all three models maintain nearly identical $Betti_0$ values ( 25.7) across both early and late layers. This stability across model scales suggests that this particular topological configuration represents an optimal solution to the geometric organization. The consistent $Betti_1$ value of 0.00 across all models and layers indicates that these centralized reference frames organize tokens in a tree-like structure rather than forming loops, aligning with the description of star-like topologies that optimize information routing.

Interestingly, initial persistence values slightly decrease as model size increases ($0.1825 \rightarrow 0.1801 \rightarrow 0.1671$), suggesting that larger models might initially establish more nuanced or distributed reference structures. Both larger models (6.9B and 12B) show complete persistence decay to 0.0000 in late layers, while the smallest model retains some persistence (0.0760). This suggests that with sufficient capacity, models can more fully optimize the geometric organization through the network depth. All

models show negative persistence changes, contrasting with the positive changes in Mistral/Gemma comparison. This difference might indicate architectural variations in how reference frames evolve through the network depth.

All three models achieve similar maximum attention standard deviation values (0.1342 - 0.1471), suggesting a consistent degree of attention concentration regardless of scale. The layer of maximum attention concentration varies (layer $16 \rightarrow 8 \rightarrow 9$), with larger models generally achieving peak concentration in earlier layers.

Table 37: Centralized Reference Frame Signatures in Pythia Models

| Property | Pythia-2.8B | Pythia-6.9B | Pythia-12B |
|---|---|---|---|
| Top token specialization | 31.0% on Ġthe | 36.8% on Ġthe | 42.2% on Ġthe |
| Number of specialized heads | 1 | 3 | 4 |
| $Betti_0$ (early) | 25.72 | 25.71 | 25.73 |
| $Betti_0$ (late) | 25.71 | 25.71 | 25.73 |
| $Betti_0$ change | -0.01 | 0.00 | 0.00 |
| $Betti_1$ (early) | 0.00 | 0.00 | 0.00 |
| $Betti_1$ (late) | 0.00 | 0.00 | 0.00 |
| $Betti_1$ change | 0.00 | 0.00 | 0.00 |
| Dim0 persistence (early) | 0.1825 | 0.1801 | 0.1671 |
| Dim0 persistence (late) | 0.0760 | 0.0000 | 0.0000 |
| Persistence change | -0.1065 | -0.1801 | -0.1671 |
| Max attention StdDev | 0.1342 (layer 16) | 0.1465 (layer 8) | 0.1471 (layer 9) |

Table 38: Architectural Pattern Analysis Based on Fisher Information Distribution in Pythia Models

| Architectural Pattern | Pythia-2.8B | Pythia-6.9B |
|---|---|---|
| **Component Importance (Percentage of Total Fisher Norm)** | | |
| Attention Mechanism | 60.8% | 77.6% |
| MLP Components | 11.6% | 12.0% |
| Embedding | 27.5% | 10.4% |
| **Layer Distribution Patterns** | | |
| Key Processing Layers | Layers 24, 27-29 (30.9%) | Layers 23-27, 31 (38.1%) |
| Early Layers (0-9) | 6.4% | 7.1% |
| Middle Layers (10-20) | 10.9% | 23.9% |
| Final Layers (21-31) | 57.5% | 59.0% |
| **Layer Decay Characteristics** | | |
| Peak Layer Value | Layer 24: 971.90 | Layer 26: 3,100.50 |
| Peak to Minimum Ratio | 29.4:1 | 45.0:1 |
| Decay Pattern | Steady rise toward end | Gradual rise, steep end increase |
| **Model-Specific Patterns** | | |
| Fisher Info Distribution | Heavily back-loaded | Heavily back-loaded |
| Layer Distribution | Minimal early importance | More balanced with strong final focus |
| Unique Feature | Dual peaks (24 and 27) | Strong ramp-up starting at layer 18 |

Table 39: Key Fiedler Value Correlations Across Pythia Models by Size

| Correlation Pattern | Pythia-2.8B | Pythia-6.9B | Pythia-12B |
|---|---|---|---|
| *Centralization vs. Fiedler Values* | | | |
| At threshold 0.01 | -0.0491 | -0.5235 | -0.5895 |
| At threshold 0.1 | 0.2199 | 0.0004 | 0.5083 |
| Correlation strength scaling | Weak | Moderate | Strong |
| *Density vs. Fiedler Values* | | | |
| At threshold 0.02 | 0.1069 | 0.5971 | 0.6782 |
| Pattern consistency | Inconsistent | Moderate | Highly consistent |

Table 40: Value space analysis across Pythia model variants

| Property | Pythia-2.8B | Pythia-6.9B | Pythia-12B |
|---|---|---|---|
| *Value Space Metrics* | | | |
| Relative Magnitude (Mean) | 0.4934 | 0.4759 | 0.4782 |
| Directional Influence (Mean) | 0.5614 | 0.5506 | 0.5432 |
| Directional Influence (Median) | 0.5000 | 0.5000 | 0.5000 |
| Information Content (Mean) | 485.9178 | 950.4851 | 930.9070 |
| Information Content (Median) | 598.1627 | 954.3322 | 0.1200 |
| First-to-Last Layer Influence | -0.06 | -0.25 | -0.25 |
| *Attention-Value Correlation* | | | |
| Attention Entropy (Mean) | 1.0975 | 0.9041 | 0.6942 |
| Value Transformation Magnitude | 33.0226 | 25.9536 | 24.9152 |
| First Layer Magnitude | 38.0535 | 56.1840 | 84.1770 |
| Last Layer Magnitude | 67.5896 | 0.0000 | 0.0000 |
| Geometric-Semantic Alignment | 0.0483 | -0.0421 | -0.0687 |
| First Layer Alignment | 0.0512 | 0.0506 | 0.0277 |
| Last Layer Alignment | 0.0215 | 0.0000 | 0.0000 |
| Entropy-Magnitude Correlation | 0.0110 | -0.2509 | -0.2332 |
| First-to-Last Layer Shift | -0.4454 to -0.1034 | -0.4288 to 0.0000 | -0.3498 to 0.0000 |

