# OpenReview forum: "What are you sinking? A geometric approach on attention sink"
_NeurIPS.cc/2025/Conference — NeurIPS 2025 spotlight_

### Official Review · Reviewer_idkt · 2025-06-28

**Clarity:** 3
**Significance:** 3
**Originality:** 3
**Rating:** 5
**Confidence:** 2

**Summary:**

This paper presents a framework for studying *attention sinks*, and often discussed but rarely
formally defined and studied property in attention-based models in which tokens corresponding to
seemingly neutral or irrelevant information attract disproportionate attention.

It studies reasons why sinks emerge, formally defines sink tokens, and classifies different
behaviors in "reference frames", which are claimed to occur as a result of learning
high-dimensionality latent spaces. Choices in the architecture such as the use of positional
information, and the specific type of positional info result in different reference frame behaviors
(centralized, bi-directional, distributed).

This framework and these findings are a great first step towards understanding attention sinks, and
to my knowledge have rarely been investigated in prior works. Sinks present a wide range of
opportunities and challenges in foundational models, from potential pruning methods for efficiency,
to efforts in robustness and more effective quantization.

**Questions:**

1. Section 3.3 suggests that the emergence of sinks is directly related to positional embeddings /
information. Does that mean that models without positional biases / embeddings do not form sinks?
Appendix A suggests that the softmax probability distribution mapping is one of the factors behind
forming of sinks, so perhaps positional information are what bias the emergence of sinks to earlier
tokens?
Could we say that sinks emerge due to softmax and positional information? This is a more interesting
finding (assuming it's true), and I think it belongs in the abstract, where it currently says sinks
are not an architectural phenomena (more on this in the next question).

2. There are many instances in the paper that claim sinks are not an architectural phenomena.
However, given that it only occurs in attention, I could argue that only architectures with
attention (primarily transformers) exhibit this behavior. This is also discussed in the paper. For
example lines 27-30:

> While prior work [...] treated attention sinks as model-specific phenomena, our geometric
> interpretation unifies these observations as alternative solutions to the same fundamental
> challenge, advancing understanding of transformer geometry.

In the end, it's all about transformers / NNs with attention. Therefore I recommend changing the
language to clarify this.


3. I may be misunderstanding this, so maybe you can point me in the right direction. Line 56 says
that the study by Moschella et al. shows that NNs with high dimensional latent spaces naturally
establish reference frames. However, after going over the paper, based on what I understand they're
proposing establishing a data-dependent coordinate system as to make it easier to compare latent
spaces, among other things.
Can you clarify this? If possible point out where Moschella et al. discusses the emergence of
reference frames?

4. Line 17:
> Removing this pattern degrades model performance

Could you please add references? I'm personally not convinced that eliminating sinks is necessarily
bad for performance, as there are many different approaches that eliminate (or mitigate) the effect.
Of course it's obvious that eliminating the tokens entirely post-training will affect quality, but
it's not clear to me why designing a reasonable architecture that is not prone to this would be
inferior in performance.

5. I have a few issues with the definition of sinks (Sec. 3, Eq. 1). I understand that this may be a
widely accepted definition, but I still take issue with it, because it is too dependent on those
thresholds, specifically $\tau$. Why is the value of the 90th percentile a good choice? Is there a
study on false positives?
If sinks are considered "context tokens that attract disproportionate attention from most or all
query tokens", then would it not make more sense to rank the attention weights for a single query
token, and instead look for context tokens that appear in the top P%?

I don't think this paper needs to discuss this, but I think it might be useful to point out that
the definition of sinks in Eq. 1 may not be the only one or the most reliable one, or if you believe
it is, it would help to cite prior works using this definition or briefly discuss why.

6. Can you comment on approaches such as Softpick or other similar modifications to the softmax
function and what could be their effect on the emergence of sinks?

**Ethical Concerns:**

["NO or VERY MINOR ethics concerns only"]

**Final Justification:**

All of my concerns were addresses by the authors, and as long as the changes that were requested by me and other reviewers are integrated, I recommend the acceptance of this paper.

**Limitations:**

yes

**Paper Formatting Concerns:**

N/A.

**Quality:**

3

**Strengths And Weaknesses:**

### Strengths

1. Formal definition of attention sinks, with reasonable arguments as to why they occur. This has
  been a rarity in papers discussing sinks directly in my experience.

2. Investigation of the effects of different types of positional information / embeddings on the
  types of reference frames and therefore sink behavior.

3. Well-written and clearly organized, and relatively easy to ready and understand.

### Weaknesses
1. The major limitation in my opinion is a lack of background references. It would certainly be
  helpful to readers in other research areas (including myself) to read prior works.
  Geometric properties such as reference frames could use more extensive definitions, or references
  for those unfamiliar (again including myself).

2. As pointed out in the paper as well, the study is limited to only certain checkpoints of the
models being studied. It would be interesting to see this study conducted throughout training. It
would also be interesting to find out whether much smaller models trained with limited resources
exhibit similar sink behavior, and that way the study could be done more efficiently.

3. (Minor) Experiments are limited to various LLMs, and not other foundation models in the visual
  or audio space. While I think having more breadth would certainly help, I don't think it's a
  necessity. More specifically, I think the findings affect those other applications as well, and
  not just LLMs, so if at all possible, I recommend looking into other foundation models beyond
  language.


### Minor Notes:

1. Multiple citations appear to be out of date or referencing the preprint instead.
  Moschella et al. "Relative representations enable zero-shot latent space communication." was published in ICLR 2023.
  Xiao et al. "Efficient streaming language models with attention sinks." was published in ICLR 2024.

2. Other works prior to Xiao et al. (2023) have also discussed the behavior of sinks without using
  the term:
  * Clark et al. "What does BERT look at? An Analysis of BERT’s Attention.": https://aclanthology.org/W19-4828.pdf.
  * Kovaleva et al. "Revealing the Dark Secrets of BERT.": https://arxiv.org/abs/1908.08593.

3. In Figure 1, could you change the color or just the visualization of connections for Centralized
  and Distributed, since they are not layer-specialized? It can be confusing as the straight line is
  labeled "Early layers connection" in the legend. Or is that intentional?

---

> ### Author Rebuttal · Authors · 2025-07-25
>
> We thank the reviewer for the accurate review.
>
> $\textbf{Regarding the "minor notes":}$
>
> - $\textsf{Out of date citations}$: Thank you for catching this. We will update all citations, including Moschella et al. (2023) and Xiao et al. (2024), to their correct, published versions.
> - $\textsf{Prior work on sink-like behavior}$: While Xiao et al. coined the term "attention sink," the underlying behavior was indeed observed in earlier foundational work on BERT interpretability. We will absolutely add citations to Clark et al. (2019) and Kovaleva et al. (2019) in our related works section to provide a more complete historical context.
> - $\textsf{Figure 1 visualization}$: Thank you for pointing out this potential confusion. The visualization was intended to show that connections in Centralized and Distributed frames are not layer-specific, unlike the Bidirectional frame. We will revise the figure and its legend to make this distinction clearer.
>
>
> $\textbf{Regarding weakness 1: lack of background references}$
>
> We will add a new section in the Appendix that provides an introduction on the key concepts from information geometry and topological data analysis.
>
>
> $\textbf{Regarding weakness 1: study on static checkpoints}$
>
> This is a key limitation that we acknowledge. A full analysis throughout the entire training process is computationally prohibitive for models of this scale. Your suggestion to study smaller models is very good and our Random Matrix Theory analysis on Pythia checkpoints (in Appendix B, due to space constraint) is a first step in this direction, showing that reference frames emerge in the earliest stages of training. We will highlight this finding more prominently and explicitly state in our limitations section that a more granular temporal study, perhaps on smaller, more efficiently trainable models, is a crucial direction for future work.
>
>
> $\textbf{Regarding weakness 1: limited Scope (LLMs only)}$
>
> Our work focuses on LLMs as this is the primary domain where attention sinks have been formally studied. However, we agree that our geometric framework is general and likely applicable to any Transformer-based architecture. We thought that this paper was rather dense in information already, so we decided to focus on text and save the other modalities for a possible future work.
>
>
> $\textbf{Regarding question 1 and 2: causality (Softmax + Positional Embeddings) and "not an architectural phenomenon"}$
>
> We agree our wording could be more precise.
>
> - Causality: You are absolutely correct. Our central argument is that sinks emerge due to the interplay between the softmax function and the inductive biases of position encoding. The softmax creates the geometric pressure for sparsity on the probability simplex (as discussed in Appendix A), and the position encoding provides the specific bias that determines where these sparse points (sinks) will form. This is indeed a core finding, and we will revise the abstract and introduction to state this two-part mechanism more explicitly.
> - "not an architectural phenomenon": Your critique of this sentence is fair, when we wrote this, we meant that sinks are not a bug or an arbitrary quirk specific to one model implementation (e.g., LLaMA vs. Mistral). Rather, they are a fundamental, predictable consequence of the core components of the Transformer architecture itself. In that sense, they are an architectural phenomenon. We will revise our language to be more precise, stating that reference frames are a fundamental geometric principle within the Transformer architecture, emerging predictably from the interaction of its core components. Thank you for pushing for this clarification.
>
> $\textbf{Regarding question 3: interpretation of Moschella et al. (2023)"}$
>
> Thank you for this very careful reading. Moschella et al. propose using relative representations to establish a shared coordinate system, they do not claim that such frames emerge naturally in standard training. Our intention was to cite their work as providing the conceptual inspiration for why reference frames are such a powerful and necessary structure in high-dimensional spaces, which motivated our search for their emergent counterparts. We will revise line 56 and related text to state that Moschella et al. demonstrated the utility and proposed the creation of such frames, which provided a key motivation for our work.
>
> $\textbf{Regarding question 3: reference for performance degradation"}$
>
> The primary reference for performance degradation upon removing the initial tokens (the attention sink) is the original paper by Xiao et al. (2024), "Efficient Streaming Language Models with Attention Sinks". They demonstrate that for streaming applications, where a fixed-size cache is used, preserving the attention sink tokens in the cache is crucial for maintaining model performance. We will add this citation explicitly at line 17.
> We also agree with your nuanced view: our claim is not that any architecture designed without sinks is inferior. Rather, simply removing the sink tokens post-hoc from a model that was trained to rely on them is detrimental. Designing an architecture from the ground up that accomplishes the same geometric stabilization without a sink is an exciting research direction. We will clarify this important distinction in the text.
>
> $\textbf{Regarding question 3: definition of sinks"}$
>
> $\textsf{On the use of thresholds and the 90th percentile}$
>
> We acknowledge the reviewer's concern that any threshold-based definition carries a degree of arbitrariness. Our approach was guided by two principles: aligning with the conceptual understanding of the phenomenon and choosing a conservative measure to ensure the robustness of our findings.
>
> - The term "attention sink" implies a token that attracts a disproportionately large share of the total attention budget. Our definition aims to capture this by looking for attention weights ($\alpha_{ij}$) that are outliers in the global distribution. The 90th percentile for $\tau$ was chosen as a standard statistical convention to isolate these extreme values, the top 10% of all attention links. This ensures we are not just flagging moderately high attention, but truly exceptional concentrations of it.
> - The role of the frequency threshold ($\gamma$): We want to emphasize that the $\tau$ threshold does not operate in isolation. The second threshold, $\gamma$ (frequency), is critical and directly addresses the reviewer's point about consistency. A token is only identified as a sink if it exceeds the high attention threshold $\tau$ for a large fraction ($\gamma \ge$ 0.3) of all source tokens. This combination ensures that we identify tokens that are not just attended to strongly by one or two queries, but that act as consistent, sequence-wide attractors. While we did not conduct a formal false-positive study, this dual-threshold requirement is designed to be highly resistant to false positives; a token must be both an outlier in attention weight and highly consistent in its role as an attractor.
>
> $\textsf{On the proposed alternative definition}$
>
> The reviewer's suggestion to rank attention weights for each query and look for tokens that consistently appear in the top P% is an excellent and valid method for identifying tokens that are consistently important to other tokens. We believe this method would be highly effective for identifying stable semantic or syntactic relationships. However, we chose our definition because it is tailored to identify a slightly different phenomenon that is central to our paper's thesis: the existence of universal geometric anchors. The reviewer's method identifies tokens that are consistently high-rank. For example, for the query "cat," the token "mouse" might consistently be in the top 5% of attended-to tokens. This captures relative importance.
>
> Our method identifies tokens that receive a high absolute quantum of attention, regardless of rank. A token could be the #1 attended-to token for every query, but if the attention distribution is very flat, its absolute weight might never cross the 90th percentile threshold. Conversely, a token like [BOS] might consistently receive 30% of the attention from every other token. It would always be highly ranked, but more importantly, it consumes a disproportionate, absolute share of the probability mass.
>
> The idea is that attention sinks function as reference points in a coordinate system. For this, we need to find the "gravitational wells" of the attention map, like the points that command a large, absolute portion of the attention budget from across the sequence. Our definition, by focusing on outlier absolute weights, is specifically designed to locate these universal anchors. The reviewer's proposed method, while very good, is better suited for analyzing pairwise token importance.
>
> We will add a note in the appendix clarifying the rationale for our definition in contrast to other valid alternatives, like the one proposed.
>
> $\textbf{Regarding question 4: effects of softmax modifications"}$
>
> Standard softmax creates the pressure for the kind of sparsity we observe in classic attention sinks. Modifying this function would not remove the fundamental need for a reference frame, but would change its implementation.
> Approaches like Softpick or sparse-softmax enforce a "harder" sparsity, often forcing the model to select a small, discrete set of tokens to attend to. This would prevent the formation of a "lazy" sink where one token passively collects a soft distribution of attention from all others. Instead, we hypothesize that these methods would force the model to implement a more dynamic reference frame. For any given query, the model would be forced to make a hard choice, selecting a small set of tokens to receive 100% of the attention. These tokens would become explicit, context-dependent reference points for that specific computational step.

---

> > ### Comment · Reviewer_idkt · 2025-08-04
> >
> > I thank the authors for responding to my review. I have no further questions, and will be updating my final review accordingly.

---

### Official Review · Reviewer_ATpd · 2025-07-01

**Clarity:** 3
**Significance:** 3
**Originality:** 3
**Rating:** 4
**Confidence:** 2

**Summary:**

This paper studies transformer "attention sinks" as geometric reference frames that establish coordinate systems in representation space. The authors identify three types: centralized (single dominant anchor), distributed (multiple weaker anchors), and bidirectional (dual anchors), each determined by position encoding choices. Using topological, spectral, and information-theoretic analysis across multiple transformer architectures, they show these reference frames emerge naturally during training as optimal solutions to geometric constraints, serving as essential infrastructure for stable token relationships rather than inefficiencies.

**Questions:**

N/A

**Ethical Concerns:**

["NO or VERY MINOR ethics concerns only"]

**Final Justification:**

I'm not an expert of this topic, but I'd like to maintain my positive score.

**Quality:**

3

**Strengths And Weaknesses:**

Strength

I'm not an expert on attention sink and related topics, but I found this explanation very interesting and novel. Understanding the attention in LLM could be very important to the field.

The paper is clear about the assumption, experiment and analysis, making it clear about the general idea without knowing the theoretical details.

In the experiment, the authors run a comprehensive cross-architecture analysis. The breadth of architectures tested demonstrates generalizability across different model families and scales.

Weakness

While the paper shows strong correlations between position encoding types and reference frame emergence, the authors don't perform controlled experiments changing only position encoding while keeping other architectural elements constant to definitively prove causation. Also, it lacks a fundamental theoretical explanation for why these specific patterns emerge.

---

> ### Author Rebuttal · Authors · 2025-07-25
>
> We thank the reviewer for the review.
>
> $\textbf{Regarding weakness 1: lack of experiments to prove causation}$
>
> We agree that a perfectly controlled experiment, training multiple large-scale models from scratch where only the position encoding scheme is varied, would provide the most definitive proof of causation. However, the huge computational cost is beyond the resources available for this work.
>
> Instead, we sought to build a strong case for a causal link through two complementary lines of evidence:
>
> - Consistency across many architectures: our study was designed to show that the link between position encoding and reference frame type is not an artifact of a single model but a consistent principle. We demonstrate that all analyzed models using standard RoPE (LLaMA 3.1/3.2, Mistral, Gemma) develop centralized frames, all models with NTK-aware scaled RoPE (Qwen 2.5, Phi-2) develop distributed frames, and all bidirectional encoder models with absolute embeddings (BERT, XLM-RoBERTa) develop bidirectional frames. The consistency of this pattern across different model families, sizes, and training datasets provides robust correlational evidence that strongly suggests the position encoding scheme is the primary causal factor.
> - A mechanistic explanation: our paper provides a direct mathematical explanation for why this causal link exists. In Section 3.4, we explain the underlying mechanism for each encoding type. For instance, we explain that standard RoPE's formulation results in an identity matrix transformation for the first token ($R_0 = I$), giving it a unique, privileged computational status that naturally leads to its emergence as a centralized reference point (lines 167-169). Similarly, we explain how the scaling factor in NTK-aware RoPE diminishes this advantage, creating the conditions for distributed frames (lines 181-184).
>
> While we did not perform the interventional experiment suggested, we believe our combination of consistent, large-scale observational evidence and a clear, mathematically-grounded causal mechanism provides a compelling argument for the claims made. We will, however, add a note to the limitations section acknowledging that this type of controlled experiment would be a valuable direction for future research.
>
>
> $\textbf{Regarding weakness 2: lack of theoretical explanation on the emergence of the phenomenon}$
>
> We respectfully believe that our paper does propose a fundamental theoretical explanation, which is rooted in the interaction between two core principles of the transformer architecture. Perhaps we can state it more directly here and ensure it is properly written in the final version.
>
> Our theoretical argument is that reference frames (and thus attention sinks) emerge as an optimal solution to a geometric problem created by the interplay of:
> - The information geometry of the softmax function: as explained in section 3.2 (and further in Appendix A), the softmax function constrains attention outputs to the probability simplex ($\Delta^{n-1}$). We argue that this is not just a normalization step but a fundamental geometric constraint. This constraint, governed by the Fisher information metric, creates a curved representational manifold that privileges sparse distributions, that is, it creates a mathematical pressure to concentrate attention probability mass onto a small number of tokens. This is the general force that necessitates the creation of "sinks."
> - The inductive bias of position encodings: while the softmax creates the pressure to form sinks, the specific mathematical structure of the position encoding scheme provides the inductive bias that determines the pattern of these sinks. Section 3.4 is dedicated to this point, providing a detailed theoretical account for each reference frame type:
> 1. Centralized frames emerge because standard RoPE's math gives a unique computational advantage to the first token.
> 2. Distributed frames emerge because scaled RoPE modifies this math to weaken the first token's advantage, allowing other tokens to serve as local anchors.
> 3. Bidirectional frames emerge because absolute position embeddings explicitly inject position markers that the encoder architecture can leverage as stable start and end anchors.
>
> So, our theory says that attention sinks are not an accident, but a necessary geometric feature. The softmax creates the "reason" (a need for sparse, stable reference points on the probability simplex), and the position encoding provides the "way" (the specific geometric biases that shape the final structure).
>
> We will revise the introduction and abstract to make this two-part theoretical argument more explicit.

---

### Official Review · Reviewer_JUDb · 2025-07-02

**Clarity:** 4
**Significance:** 3
**Originality:** 3
**Rating:** 5
**Confidence:** 3

**Summary:**

The paper presents a geometric interpretation of the attention sink phenomenon in transformer models, arguing that attention sinks are not merely architectural artifacts but manifestations of fundamental geometric principles related to the establishment of reference frames in high-dimensional spaces. The authors identify three distinct types of reference frames—centralized, distributed, and bidirectional—that correlate with the attention sink phenomenon and emerge as optimal solutions for establishing stable coordinate systems during training. They analyze how architectural components, particularly position encoding implementations, influence the specific type of reference frame that emerges. The paper's contributions include unifying diverse attention patterns across architectures under a single geometric principle, providing insights into how architectural choices affect attention organization, and laying the foundation for deliberate reference frame engineering in transformer models.

**Questions:**

1. You have provided extensive analysis on several model families (LLaMA, Qwen, BERT variants, etc.) at various parameter scales. However, most analyses focused on models up to ~12B parameters. Could you discuss whether and how your findings (e.g., centralized, distributed, bidirectional reference frames) scale to significantly larger models (70B+ parameters, or GPT-class models)?
2. Your analysis provides strong theoretical and empirical insights into attention sinks. Can you elaborate more concretely on how these insights might directly inform architectural choices or training strategies in practice?
3. Your paper utilizes advanced methodologies (persistent homology, spectral graph theory, information geometry) extensively. Could you clarify why each analysis was necessary to support your core conclusions? Could any of these analyses be considered redundant or confirmatory?
4. You briefly suggest in your conclusion the intriguing idea of "reference frame engineering" using attention sinks. Can you expand more explicitly on potential methods or experiments for deliberately manipulating attention sinks to achieve desired model behaviors?

**Ethical Concerns:**

["NO or VERY MINOR ethics concerns only"]

**Final Justification:**

Resolved Issues:
1. Complexity and Necessity of Analyses: The authors clarified that the use of diverse analytical tools (persistent homology, spectral graph theory, information geometry, etc.) was not redundant but complementary, each answering a specific research question. They committed to revising the methodology section to clearly map each method to its respective insight, which significantly improves accessibility.
2. Lack of Practical Implications: The rebuttal provides specific pathways for using reference frame insights in architectural decisions.
3. Scalability to Larger Models: While empirical validation on >12B models is missing due to compute limits, the authors provide a clear theoretical rationale for scale-invariant behavior of reference frames, based on architectural primitives (e.g., softmax constraints, RoPE properties). They acknowledge this limitation and appropriately classify scalability as a testable future hypothesis rather than an overclaim.
Remaining (Minor) Concerns:
1. The lack of large-scale empirical validation (e.g., 70B+ models) limits the immediate practical generalizability, though theoretical support is persuasive.
Despite a few open questions (e.g., scale-up validation), the paper is methodologically sound, conceptually rich, and forward-looking. The rebuttal was detailed, thoughtful, and addressed concerns directly. Given the novelty, depth, and clarity of response, I believe this paper makes a meaningful and lasting contribution to understanding transformer architectures and deserves a spot at NeurIPS.

**Limitations:**

Yes.

**Paper Formatting Concerns:**

None.

**Quality:**

4

**Strengths And Weaknesses:**

Strengths:
1. The submission is technically robust and thorough. It provides a well-founded theoretical explanation for attention sinks, rooted in geometry and information theory.
2. The paper’s empirical analysis is impressively comprehensive: it spans multiple model families and sizes, and uses complementary analysis techniques to validate each claim.
3. The submission is generally well-written and organized.
4. This work advances the mechanistic understanding of transformers by reframing attention sinks as necessary geometric structures rather than model artifacts.

Weaknesses:
1. One minor concern is that some of the analysis, while enlightening, is quite complex (e.g. persistent homology on attention graphs, information geometry formulations). This raises the question of whether all these analyses were necessary to support the core claims or if a subset would suffice – however, the inclusion of multiple perspectives does strengthen confidence in the findings.
2. Although the analysis covers various transformer models and scales (up to ~12B parameters), generalizability to significantly larger models (e.g., 70B+ parameters) remains unclear. Clarifying theoretical expectations or providing additional empirical insights regarding larger-scale models would further strengthen the paper’s impact.
3. The contribution primarily centers on theoretical understanding and conceptual insights rather than immediate practical improvements to transformer architectures or their performance metrics.
4. The paper’s density of content is quite high. Readers who are not familiar with certain fields (e.g., topological data analysis or information geometry) might find parts of the presentation difficult to follow. A bit more gentle explanation or an appendix tutorial could help a broader audience.

---

> ### Author Rebuttal · Authors · 2025-07-25
>
> We thank the reviewer for the accurate review.
>
> $\textbf{Regarding weakness 1: complexity and necessity of the analyses}$
>
> Our goal was to show that the evidence for reference frames is not an artifact of one particular analytical lens, but a consistent signal across topological, spectral, information-theoretic, and vector-space perspectives. In our revision, as detailed in our answer to Question 3, we will improve the methodology section to more clearly motivate why each tool was necessary to answer a specific research question, thereby guiding the reader through our analytical narrative.
>
>
> $\textbf{Regarding weakness 2: scalability to larger models}$
>
> Our analysis was computationally constrained to models up to ~12B parameters. However, as we elaborate in our answer to Question 1, our core theoretical argument is based on architectural "primitives" (softmax and position encodings) that are scale-invariant. We therefore hypothesize that our findings will generalize, and perhaps even become more pronounced, in larger models. We will explicitly state this in the limitation section.
>
> $\textbf{Regarding weakness 3: theoretical vs. practical contributions}$
>
>  Our paper moves the conversation about attention sinks from "bug" to "feature," opening up new avenues for deliberate architectural design and training strategies. We outline several concrete examples of how this theoretical insight can translate into practical applications in our answers to Question 2 and Question 4.
>
> $\textbf{Regarding weakness 4: the density of the presentation}$
>
> We agree that the paper's density could be a barrier. We will add more explanatory text to contextualize the results in our tables (as discussed in our response to Reviewer uQoB) and restructure Section 4 to better motivate our methodology. We will also add a section to the appendix to provide more background on the key concepts from topological data analysis and information geometry.
>
> $\textbf{Regarding question 1: scaling on bigger models}$
>
> Unfortunately for this project we didn’t have the resources to test on models of that size. But, while our direct empirical analysis was computationally constrained to models up to ~12B parameters, we have theoretical reasons to believe our findings will generalize to much larger models.
>
> - Our core thesis is that reference frames are not an emergent property of scale, but a fundamental consequence of architectural primitives. The geometric principles we identify, the simplex constraint of softmax and the mathematical properties of position encodings, are intrinsic to the architecture, regardless of parameter count. A 70B LLaMA-family model still uses standard RoPE, which mathematically privileges the first token. Therefore, we hypothesize it will continue to exhibit a centralized reference frame. In fact, as model capacity and the complexity of the learned representations increase, the need for a stable, efficient coordinate system to anchor the high-dimensional manifold likely becomes more critical, not less.
> - Empirical observations from the community regarding larger models appear to support this. For instance, strong, persistent attention to the [BOS] token is a widely noted phenomenon in very large decoder-only models. Our framework provides a formal explanation for these observations.
>
> So, we acknowledge that direct empirical validation on 70B+ models is an important next step. We will add to the paper that while our theory predicts scalability, this remains a hypothesis to be confirmed by future work with greater computational resources.
>
> $\textbf{Regarding question 2: how the knoledge about the reference frame could be used in practice}$
>
> Our findings suggest that the choice of position encoding is, in effect, a choice of geometric inductive bias. This allows for more principled architecture design. For example:
> - For tasks requiring robust long-range dependencies anchored to a consistent context (e.g., summarization, few-shot prompting), an architecture promoting a centralized frame (like standard RoPE) may be optimal due to its efficient, hub-and-spoke information routing.
> - For tasks requiring high contextual flexibility and sensitivity to local syntax (e.g., creative writing, complex reasoning), an architecture promoting a distributed frame (like NTK-scaled RoPE) might be better, as it creates a more adaptable, locally-aware coordinate system. This allows designers to match the geometric properties of the model to the geometric properties of the task.
>
> Knowing that models dedicate significant capacity to forming reference frames (as shown by our Fisher Information analysis), we could devise strategies to accelerate this process. For example, a pre-training or warm-up phase could use a curriculum or a specific regularization term to explicitly encourage the formation of a stable reference frame, potentially leading to faster convergence on the primary task.
>
> $\textbf{Regarding question 3: is the analysis redundant?}$
>
> We view our analysis not as redundant, but as providing a form of cross-validation, where each method offers a unique and necessary piece of the puzzle. The complexity arises from the need to build a robust case from different angles.
>
> Here is a breakdown of why each component was essential:
>
> - Topological Analysis (Persistent Homology): This was necessary to understand the high-level shape of the attention graph. It answered the question: "What is the global connectivity structure?" For example, the discovery of high $Betti_1$ values (loops) in early layers was a unique signature that immediately distinguished bidirectional frames from the star-like ($Betti_0 \approx$ 0) structures of decoders.
> - Spectral Graph Theory: This analysis quantified the properties of the connectivity structures revealed by topology. It answered: "How is information concentrated and distributed within that shape?". Metrics like the Fiedler value and degree centralization allowed us to quantitatively differentiate between centralized (single dominant node) and distributed (multiple smaller nodes) frames. The sign-flipping correlation pattern we identified became a key quantitative signature.
> - Information Geometry (KL & Fisher): This moved beyond structure to function. It answered: "How critical are these reference points for the model's information processing?" The KL reduction experiment directly measured the functional cost of removing sinks, proving their importance. The Fisher information analysis revealed where in the network (i.e., which layers) the model invests its learning capacity to build these frames, highlighting the early-layer focus of centralized frames versus the mid-layer focus of bidirectional ones.
> - Value Space Analysis: This final piece connected attention patterns to their ultimate purpose: geometric transformation. It answered: "How do these reference points actually orient other tokens in the value space?" The directional influence metric was crucial here, showing that reference tokens act as coordinate axes that actively guide other representations, rather than just passively aggregating information.
> - Random Matrix Theory Analysis (in the Appendix): it clarified that the reference frames emerge in the first stages of the training, showing that the reference frames aren't just an artifact, but a useful feature to achieve proper processing.
>
> We will, however, take the reviewer's feedback to heart and revise the methodology section (section 4) to more clearly state the unique question each analysis answers, and better justifying the inclusion of each and guiding the reader through the logic.
>
> $\textbf{Regarding question 4: potential methods or experiments for "reference frame engineering"}$
>
> Building on our answer to Question 2, we can envision several concrete experimental ideas:
> - Explicit anchor tokens: it could be possible to introduce new, non-semantic "anchor" tokens into the vocabulary. During training, a regularization term could be added to the loss function to explicitly encourage (or discourage) attention to these tokens, effectively creating programmable reference points. An experiment could test if a model can be trained to use [ANCHOR_1] as a centralized sink and [ANCHOR_2] as a secondary, distributed sink.
> - Attention head specialization: Our analysis shows that specific heads specialize in attending to reference points. it could be possible to perform experiments where, during fine-tuning, these specialized heads are either frozen (to preserve the original geometric structure) or selectively pruned/retrained (to force the model to adopt a new geometric strategy). This could be a powerful tool for efficient task adaptation.
>
> Once again, we thank the reviewer for their deep engagement with our work. Their feedback has been valuable in helping us clarify our contributions and strengthen the paper.

---

### Official Review · Reviewer_uQoB · 2025-07-02

**Clarity:** 3
**Significance:** 4
**Originality:** 4
**Rating:** 5
**Confidence:** 3

**Summary:**

This paper introduces a geometric interpretation for attention sinks in transformers, proposing that these sinks emerge as optimal 'reference frames' under softmax constraints in an emergent coordinate system. Crucially, they emphasize that these are not accidental artifacts but fundamental outcomes of geometric constraints on the learning dynamics of transformers. The authors identify three reference-frame types—centralized, distributed, and bidirectional—linked to inductive biases from positional-encoding schemes, and validate these patterns across 10+ model checkpoints (e.g. LLaMA, Qwen, BERT) using topological, spectral, and information-theoretic analyses

**Questions:**

### Questions or Suggestions
- The authors state that reference frames (and the specific reference frame types they observe) are *optimal solutions* under the softmax constraint to "establishing stable coordinate systems in high-dimensional spaces". In what sense do they mean "optimal" here, optimal with respect to which exact objective? Could the authors elaborate a little bit on this.
- Could you clarify the definition of a specialized head is in Table 1? I'm trying to relate the dataset size to these measures, e.g. does it have to meet the reference frame attention threshold for all samples in the dataset?
- What precisely is "Token Specialization" in Table 1?
	- e.g. What does 100% on BOS token mean?  I assumed it meant that all specialized heads (ones that demonstrate attention sinks) only attend to the BOS.
	- What about the *65.4% on ","*, for the distributed frame? What do the other specialized heads attend to? Are these commas in the same position (absolute or relative--i.e. the first comma in the sentence) across all sentences, or different.
- As mentioned in the weaknesses section, could you provide charts of the distribution of attention sink patterns for each reference frame type across  the sentences tested (e.g. what tokens end up as attention sinks across all 500 sentences and what percentage of specialized heads focus on them, how many attention sink tokens per sentence, how many specialized heads etc) . I'm particularly interested to see how distributed among different tokens the "distributed reference frame" really is.
- How were the thresholds for what constitutes an attention sink (Section 3) set?

**Ethical Concerns:**

["NO or VERY MINOR ethics concerns only"]

**Final Justification:**

I think this is a quite interesting paper providing a lens on the attention sink phenomenon from a number of different analytical angles while providing a theoretical motivation for why these sinks appear. While the analysis is dense I think it gives a path for future work to build upon its findings to, for example, explore the necessity or relative advantages/disadvantage of attention sink formation as an architectural component.

With regards to the rebuttal, my questions were primarily around clarity of presentation and elaboration of a few definitions in the paper and the authors have responded adequately.

**Limitations:**

The authors include a brief limitations section. An extra limitation that readers find helpful is that the KL analysis on ablated sink tokens may not translate to impact on downstream tasks.

**Quality:**

3

**Strengths And Weaknesses:**

### Strengths

- The geometric analysis of attention sinks in transformer models presents a novel framing of these sinks as sites for an emergent coordinate system. Providing insight into what is going on at these tokens and why they emerge.
- The experimental setup used is thorough, while the domain of sentences used to is somewhat narrow (i.e. just *stem-oriented* sentences), multiple architectures and models are tested in a comprehensive manner.
- The conceptual framing of the paper is well presented (Sections 1-3) and the optimality perspective is interesting.
- The authors bring to bear a rich set of analytic tools such as persistent homology, spectral graph metrics, KL/Fisher information to investigate their hypotheses.


### Weaknesses
- **Sections 4-5 seems to be overly terse and a reads a bit like an information dump.** The authors introduce numerous analytic tools (persistent homology, Betti numbers, Ripser algorithm, persistence values, spectral properties, algebraic connectivity, star likeness, Gini coefficient, degree centralization, information theoretic measures, Fisher information matrix etc.) in a short span without guiding the reader to what question each tool is geared towards answering. It might be more helpful if these were presented in sections answering particular research questions / showing evidence for particular hypotheses, or at least some schematic of tool to information inferred.
- **Metric values are not contextualized for readers:** in the various result tables there are many different metrics being presented without much contextualization of why these particular numbers imply the conclusions drawn. For example, Table 1 reports Betti₁ values but does not explain what constitutes a “high” vs. “low” Betti₁ or why changes across layers are meaningful.
- **Some simple baseline analyses are missing:**
	- While it is natural that some background in the subfields used for analysis would be needed to fully understand the results, it would be helpful to start with some simpler statistical analyses that demonstrate evidence for the reference types found and then extend to the more in-depth analytic methods (potentially leaving some entirely for the appendix given the space constraints).
	- For example an initial visualization/table of the distribution of attention with respect to the sinks for each reference frame type across all the sentences tested (e.g. percentage of attention on sinks in specialized and possibly non-specialized heads, number of sinks per sentence). This is summarized in the "Token specialization" line in Table 1. But I would have expected to have seen that broken out in a bit more detail as initial evidence for the attention patterns described in prior sections.

---

> ### Author Rebuttal · Authors · 2025-07-25
>
> We thank the reviewer for the accurate review.
>
> $\textbf{Regarding weakness 1 and 3: info dump and missing simple baselines}$
>
> We agree with the reviewer's opinion. In our effort to be comprehensive, the presentation in sections 4 and 5 became overly dense.
>
> Our Plan for Revision:
>
> We will revise Section 4 (Methodology) to explicitly frame our analysis around a series of research questions. This will guide the reader and justify why each tool was necessary, transforming it from an "information dump" into a clear investigative path. The structure will be:
>
> - RQ1: What is the global shape of the attention graph? -> Tool: Topological Analysis (Persistent Homology, Betti numbers).
> - RQ2: How is information concentrated within that shape? -> Tool: Spectral Graph Theory (Fiedler value, centralization).
> - RQ3: How functionally critical are these structures? -> Tool: Information Geometry (KL Divergence, Fisher Information).
> - RQ4: How do these structures orient representations in value space? -> Tool: Value Space Analysis (Directional influence, etc.).
>
> As suggested, we will add a new subsection in Appendix (due to space constraints) dedicated to a more detailed, foundational analysis of attention sink patterns. This new section will include the requested visualizations, showing the distribution of attention sinks for each reference frame type across our test dataset.
>
>
> $\textbf{Regarding weakness 2: metric values are not contextualized for readers.}$
>
> This is a fair and important criticism. We thank the reviewer for pointing this out. To stick to page limits, the tables in the main body of the paper present a high-level summary of our findings. We would like to note that the complete, detailed tables for all experiments are provided in the appendix for full transparency.
>
> That said, we agree that the summarized values in the main text must be better contextualized to be interpretable. We will revise Section 5 to ensure that for each value presented, we explicitly explain its significance, for example what constitutes a "high" versus "low" value and why the observed numbers support our conclusions.
>
> Example for Table 1: When discussing $Betti_1$ values, we will add a clarification: "A $Betti_1$ value near zero, as seen in Llama-3.2, indicates a tree-like or star-like graph with very few cycles, which is the topological signature of a centralized structure. In contrast, the significantly higher initial Betti₁ of 19.69 for XLM-RoBERTa indicates a highly complex graph with numerous loops, a distinctive feature of the initial layer connectivity in bidirectional frames." We will apply this principle to all key metrics in the main text.
>
> $\textbf{Regarding Question 1: what do you mean by "optimal"? - definition of a "specialized head"}$
>
> By "optimal," we do not mean a globally optimal solution in a mathematical sense, but rather an emergent optimum found by the model during training via gradient descent.
>
> The objective being optimized is the standard cross-entropy loss, which, as we argue in Section 3.2 (citing Tishby et al., 2000), can be viewed through the lens of the information bottleneck principle (Equation 7). The "optimality" of reference frames arises from the interplay of two things:
>
> - The softmax constraint: This pushes attention distributions towards sparsity on the probability simplex to efficiently compress information.
> - The positional encoding bias: This creates a specific "loss landscape" where certain tokens are computationally cheaper to use as anchors.
>
> A reference frame is "optimal" in the sense that it is the most efficient strategy the model discovers to satisfy the need for a stable, sparse coordinate system (from the softmax) within the specific geometric constraints imposed by its architecture (the position encoding). We will revise the abstract and introduction to state this more precisely.
>
> A head is defined as "specialized" if it has learnt to consistently form attention sinks across our dataset. So, we classify a head as specialized if, for a high percentage of the test samples (for example, >80%), it produces an attention pattern that meets the $\textsf{sink}(j)$ definition from Equation 1. This ensures we are analyzing heads that have adopted a stable, structural role rather than ones that only occasionally exhibit sink-like behavior.
>
> $\textbf{Regarding question 2: meaning of "Token Specialization".}$
>
> This metric in Table 1 describes the behavior of the specialized heads we identified.
>
> - "100% on BOS token" for a centralized frame means that every single one of the specialized heads in that model uses the Beginning-of-Sequence token as its primary, consistent attention sink.
>
> - "65.4% on ','" for the distributed frame means that among that model's specialized heads, the comma token is the most common anchor, accounting for 65.4% of their observed sink behavior. The remaining percentage is distributed among other tokens that serve as secondary anchors, such as articles ("the", "a") or periods.
>
> It's interesting to note that in the second case the "special tokens" are function-dependent, not position-dependent. A comma acts as a reference point because of its syntactic role, regardless of where it appears in the sentence.
>
> We are gonna add a proper explanation regarding this point in the paper.
>
> $\textbf{Regarding question 3: extra charts for attention sink distribution.}$
>
> We agree that providing these visualizations would greatly strengthen the paper and make the concept of a distributed frame much clearer. We plan to add a new section (probably in the appendix) dedicated to this. It will include charts showing the distribution of tokens that become sinks for each reference frame type across our dataset.
>
> $\textbf{Regarding question 4: how thresholds were set.}$
>
> The thresholds $\tau$ (attention weight) and $\gamma$ (frequency) in Equation 1 were set empirically to be conservative and robust.
>
> - $\tau$ (90th percentile): This was chosen as a standard statistical convention to isolate only the most extreme, outlier attention weights, ensuring we focus on truly disproportionate attention.
>
> - $\gamma$ (0.3-0.5): This frequency threshold ensures that we only identify tokens that are consistent attractors across a large portion of the dataset, filtering out anomalies that might appear in only a few samples.
>
> Our method identifies tokens that receive a high absolute quantum of attention, regardless of rank. A token could be the #1 attended-to token for every query, but if the attention distribution is very flat, its absolute weight might never cross the 90th percentile threshold. Conversely, a token like [BOS] might consistently receive 30% of the attention from every other token. It would always be highly ranked, but more importantly, it consumes a disproportionate, absolute share of the probability mass.
> The idea is that attention sinks function as reference points in a coordinate system. For this, we need to find the "gravitational wells" of the attention map, like the points that command a large, absolute portion of the attention budget from across the sequence. Our definition, by focusing on outlier absolute weights, is designed to locate these anchors.

---

> > ### Comment · Reviewer_uQoB · 2025-08-06
> >
> > Thanks for your response, I think all my questions have been answered and I see a few common themes across the reviews so I'm confident that the clarity of the paper will increase. I'll update (increase) my score accordingly.

---

### Note · Authors · 2025-08-15

We thank the area chair and all reviewers for their feedback.
We recognize the primary concerns centered on the paper's density and the need to more sharply define our core theoretical contribution.

Our central thesis, which we will state more explicitly in the revision, is that reference frames are not an accidental artifact but an optimal geometric solution emerging from the interplay of two core architectural principles:

- The "Why": The information geometry of the softmax function creates a fundamental pressure for sparse, stable anchor points on the probability simplex. This is the general force that necessitates the creation of "sinks."

- The "How": The specific mathematical inductive bias of the position encoding scheme (e.g., standard RoPE vs. NTK-scaled RoPE) determines the resulting pattern of these anchors, leading to centralized, distributed, or bidirectional frames.

This two-part mechanism provides a causal explanation for the consistent patterns we observed across numerous models. While a fully controlled interventional experiment was beyond our resources, our argument is strongly supported by this clear mechanistic link combined with extensive observational evidence.

To address the paper's density and improve its narrative flow, we will implement a significant revision focused on clarity:

- Reframing: We will imrpove the methodology (Section 4) making the research questions explicit. This will transform it from a dense "information dump" into a clear investigative path, motivating why each analytical tool (topological, spectral, etc.) was essential to uncover a different facet of the phenomenon.

- Interpretation: We will revise the results section (Section 5) to add explicit context for the values in our tables, explaining their significance (e.g., what constitutes a "high" vs. "low" value) to make our findings directly interpretable (the full tables are still available in the appendix).

- Supporting material: We will add appendices to provide background on key technical concepts and include the additional visualizations requested by reviewers.

We believe these revisions will improve the paper's clarity, and better highlight how this new geometric understanding of attention opens concrete avenues for principled architecture design, training optimization, and model control.

---

### Decision · Program_Chairs · 2025-09-17

**Decision:**

Accept (spotlight)

**Comment:**

This paper advances a clear, novel geometric account of attention sinks as emergent reference frames (centralized, distributed, bidirectional) shaped by softmax geometry and positional encodings. Evidence spans multiple architectures and checkpoints (e.g., LLaMA/Qwen/BERT) with complementary analyses (topological, spectral, information-theoretic). Reviewers are positive (two “Accept,” one “Borderline-Accept”), citing strong conceptual contribution and thorough cross-architecture validation; residual concerns (presentation density, absence of fully controlled causality experiments, very-large-scale validation) are primarily expository and scope-related. The rebuttal directly addresses these with concrete camera-ready commitments: restructuring methodology around explicit research questions, contextualizing metrics, adding background/visualizations, clarifying “optimality” and sink definitions, and precisely situating claims (softmax + positional encodings). Given the strength and the reviewers’ post-rebuttal endorsements, the AC decides to accept the submission.